# 📏RULER: What's the Real Context Size of Your Long-Context Language Models?

**Cheng-Ping Hsieh**[*], **Simeng Sun**[*], **Samuel Kriman, Shantanu Acharya**
**Dima Rekesh, Fei Jia, Yang Zhang, Boris Ginsburg**

NVIDIA
{chsieh,simengs}@nvidia.com

## Abstract

The needle-in-a-haystack (NIAH) test, which examines the ability to retrieve a piece of information (the "needle") from long distractor texts (the "haystack"), has been widely adopted to evaluate long-context language models (LMs). However, this simple retrieval-based test is indicative of only a superficial form of long-context understanding. To provide a more comprehensive evaluation of long-context LMs, we create a new synthetic benchmark RULER with flexible configurations for customized sequence length and task complexity. RULER expands upon the vanilla NIAH test to encompass variations with diverse types and quantities of needles. Moreover, RULER introduces new task categories *multi-hop tracing* and *aggregation* to test behaviors beyond searching from context. We evaluate 17 long-context LMs with 13 representative tasks in RULER. Despite achieving nearly perfect accuracy in the vanilla NIAH test, almost all models exhibit large performance drops as the context length increases. While these models all claim context sizes of 32K tokens or greater, only half of them can maintain satisfactory performance at the length of 32K. Our analysis of Yi-34B, which supports context length of 200K, reveals large room for improvement as we increase input length and task complexity. We open source RULER to spur comprehensive evaluation of long-context LMs.

## 1 Introduction

Recent advancements in AI system engineering (Dao et al., 2022; Jacobs et al., 2023; Fu et al., 2024) and language model designs (Chen et al., 2023; Xiong et al., 2023) have enabled efficient scaling up of context length for language models (Liu et al., 2024a; Young et al., 2024). Previous works (AI21, 2024; X.AI, 2024; Reid et al., 2024; Anthropic, 2024) commonly adopt synthetic tasks, such as passkey retrieval (Mohtashami & Jaggi, 2023) and needle-in-a-haystack (Kamradt, 2023) to evaluate long-context LMs. However, these evaluations are used inconsistently across works and reveal merely the retrieval capability, failing to gauge other forms of long-context understanding.

In this work, we propose RULER, a new benchmark to evaluate long-context modeling capabilities for language models. RULER contains four task categories to test behaviors (Ribeiro et al., 2020) beyond simple retrieval from context:

1. **Retrieval:** we extend the needle-in-a-haystack (Kamradt, 2023, NIAH) test to evaluate retrieval capability with diverse types and quantities of needles.

2. **Multi-hop Tracing:** we propose *variable tracking*, a minimal proxy task for coreference chain resolution to check the behavior of tracing entities with multi-hop connections.

3. **Aggregation:** we propose *common/frequent words extraction*, proxy tasks for summarization to test the ability to aggregate relevant information that spans long-range context.

---

[*] Authors contributed equally.

| Benchmark & Task | Avg Len | Type | Diverse Tasks | Min. Parametric Knowledge | Controllable Context |
|---|---|---|---|---|---|
| **ZeroSCROLLS** | ∼10k | realistic | ✓ | ✗ | ✗ |
| **L-Eval** | ∼8k | realistic | ✓ | ✗ | ✗ |
| **BAMBOO** | ∼16k | realistic | ✓ | ✓ | ✗ |
| **LongBench** | ∼8k | hybrid | ✓ | ✗ | ✗ |
| **LooGLE** | ∼20k | hybrid | ✓ | ✓ | ✗ |
| **InfiniteBench** | ∼200k | hybrid | ✓ | ✓ | ✗ |
| **Needle-in-a-haystack (NIAH)** | any | synthetic | ✗ | ✓ | ✓ |
| **Passkey / Line / KV Retrieval** | any | synthetic | ✗ | ✓ | ✓ |
| **RULER (Ours)** | any | synthetic | ✓ | ✓ | ✓ |

Table 1: Comparison between existing long-context benchmarks and RULER. "Realistic" type refers to human-annotated while "synthetic" type refers to auto-generated. RULER includes diverse task domains beyond retrieval, reduces reliance on parametric knowledge with synthetic input, and offers flexibility to control the contexts for different sequence lengths and task complexities. In RULER, contexts can be adjusted by changing the volume or placement of relevant and distracted information.

4. **Question Answering:** we add distracting information to the input of existing short-context QA datasets to evaluate question answering capability at various context sizes.

Compared to existing realistic benchmarks (Table 1), RULER consists solely of synthetic tasks, which offer the flexibility to control sequence length and task complexity. The synthetic input in RULER reduces reliance on parametric knowledge, which interferes with the utilization of long-context input in realistic tasks (Shaham et al., 2023; Bai et al., 2023).

Using RULER, we benchmark Gemini-1.5 (Reid et al., 2024), GPT-4 (OpenAI: Josh Achiam et al., 2023), and 15 open-source models with context length ranging from 4k to 128k. Despite achieving nearly perfect performance on the vanilla NIAH test, almost all models exhibit large degradation on more complex tasks in RULER as sequence length increases. While all models claim context size of 32k tokens or greater, our results indicate that only half of them can effectively handle sequence length of 32K by exceeding a qualitative threshold. Moreover, almost all models fall below the threshold before reaching the claimed context lengths. To obtain fine-grained model comparisons, we aggregate performance from 4k to 128k with two weighted average scores where the weights simulate the length distribution of real-world use cases. The top two models - Gemini-1.5 and GPT-4, consistently outperform other models regardless of the chosen weighting scheme.

We further analyze Yi-34B, which claims context length of 200K and achieves reasonably good performance on RULER among open-source models. Our results demonstrate large degradation in Yi's performance as we increase input length and task complexity. At large context sizes, Yi-34B often returns incomplete answers and fails to precisely locate the relevant information. Furthermore, we observe two behaviors emerging with the scaling of context size across multiple models: the increased reliance on parametric knowledge and the increased tendency to copy from context for non-retrieval tasks. Our additional ablations demonstrate that training on longer sequences does not always lead to better performance on RULER, and that larger model sizes positively correlate with better long-context capabilities. Finally, we show that non-Transformer architectures, such as RWKV and Mamba, still lag behind Transformer by large margins on RULER.

Our contributions are as follows:

- We propose a new benchmark RULER for evaluating long-context language models via synthetic tasks with flexible configurations.
- We introduce new task categories, specifically multi-hop tracing and aggregation, to test behaviors other than retrieval from long context.
- We evaluate 17 long-context LMs using RULER and perform analysis across models and task complexities.

We open source RULER to spur future research in long-context language models.[1]

---

[1] https://github.com/hsiehjackson/RULER

## 2 Related Work

**Long-context Language Models.** Numerous long-context language models have been introduced lately owing to the progress in engineering, architectural, and algorithmic designs. Flash attention (Dao et al., 2022; Dao, 2023) and Ring attention (Liu et al., 2023) significantly reduce the memory footprint required for processing long context. Various sparse attention mechanisms (Child et al., 2019; Jaszczur et al., 2021) such as shifted sparse attention (Chen et al., 2024), dilated attention (Ding et al., 2023), and attention sinks (Han et al., 2023; Xiao et al., 2024b) were employed to enable efficient context scaling. Novel position embedding methods were proposed to improve length extrapolation in Transformers (Vaswani et al., 2017), including ALiBi (Press et al., 2022), xPOS (Sun et al., 2023b), and RoPE (Su et al., 2023) variants (Chen et al., 2023; Xiong et al., 2023; Peng et al., 2024; Liu et al., 2024b; Ding et al., 2024; Zhu et al., 2024). Another line of research focuses on reducing context size. This can be achieved by caching previous context using recurrence mechanism (Zhang et al., 2024a; Bulatov et al., 2023; Martins et al., 2022; Wu et al., 2022), retrieving relevant information from context (Xu et al., 2024a; Mohtashami & Jaggi, 2023; Wang et al., 2024; Tworkowski et al., 2024; Xiao et al., 2024a), or preserving the salient information via compression (Jiang et al., 2023). Finally, novel architectures (Gu et al., 2022; Fu et al., 2023a; Poli et al., 2023; Fu et al., 2023b; Sun et al., 2023a; Beck et al., 2024; Sun et al., 2024) such as Mamba (Gu & Dao, 2023) and RWKV (Peng et al., 2023) have also been proposed to efficiently handle long-context input.

**Long-context Benchmarks and Tasks.** Our work is closely related to other works on benchmarking long-context language models. ZeroSCROLLS (Shaham et al., 2023) covers ten realistic natural language tasks, such as long-document QA and (query-based) summarization. L-Eval (An et al., 2024) also uses realistic data, which was filtered manually to ensure quality. LongBench (Bai et al., 2023) contains tasks in a bilingual setting. InfiniteBench (Zhang et al., 2024b) includes tasks with length greater than 100K tokens. LTM (Castillo et al., 2024) targets the evaluation of long-term conversations. To isolate the effect of parametric knowledge, previous works (Dong et al., 2023; Li et al., 2023b) also propose to use documents posted online later than a certain cutoff date, or leverage extremely low-resource materials (Tanzer et al., 2024). Compared to realistic benchmarks, synthetic tasks are more flexible to control the setup (e.g., sequence length and task complexity) and less affected by parametric knowledge. Recent works have primarily focused on retrieval-based synthetic tasks (Kamradt, 2023; Mohtashami & Jaggi, 2023; Li et al., 2023a; Liu et al., 2024d; Lee et al., 2024), with a few investigate other aspects, including fact reasoning (Kuratov et al., 2024; Karpinska et al., 2024), long-range discourse modeling (Sun et al., 2022), question answering (Levy et al., 2024; Yuan et al., 2024), many-shot in-context learning (Agarwal et al., 2024; Bertsch et al., 2024; Xu et al., 2024b), and code understanding (Liu et al., 2024c).

## 3 The RULER Benchmark

RULER comprises tasks across four categories: *retrieval*, *multi-hop tracing*, *aggregation*, and *question answering*. Evaluation examples in RULER are automatically generated based on input configurations (see Table 2) that define the length and complexity of each input. Within a constrained domain as in RULER, the task complexity can be thought of as a function of the number of target output tokens and the signal-to-noise ratio in the context. We point readers to (Goldman et al., 2024) for more comprehensive discussion on evaluation task design for long-context language models.

### 3.1 Retrieval: Needle-in-a-haystack (NIAH)

Recent works (Reid et al., 2024; Anthropic, 2023) commonly employ the needle-in-a-haystack (Kamradt, 2023, NIAH) test to evaluate long-context modeling capability. The NIAH test is reminiscent of the extensively studied (Hopfield, 1982; Graves et al., 2014; Olsson et al., 2022; Arora et al., 2024) *associative recall* tasks, in which relevant information needs to be retrieved from context given a sufficient query. In RULER, we include multiple retrieval-based tasks, extending the vanilla NIAH test to evaluate models based on three

| Task | Configuration | Example |
|------|---------------|---------|
| Single NIAH (S-NIAH) | type_key = word
type_value = number
type_haystack = essay
size_haystack ∝ context length | (essays) ......
One of the special magic numbers for long-context is: 12345. ......
What is the special magic number for long-context mentioned in the provided text?
Answer: 12345 |
| Multi-keys NIAH (MK-NIAH) | num_keys = 2
type_key = word
type_value = number
type_haystack = essay
size_haystack ∝ context length | (essays) ......
One of the special magic numbers for long-context is: 12345.
One of the special magic numbers for large-model is: 54321.
......
What is the special magic number for long-context mentioned in the provided text?
Answer: 12345 |
| Multi-values NIAH (MV-NIAH) | num_values = 2
type_key = word
type_value = number
type_haystack = essay
size_haystack ∝ context length | (essays) ......
One of the special magic numbers for long-context is: 12345.
One of the special magic numbers for long-context is: 54321.
......
What are all the special magic numbers for long-context mentioned in the provided text?
Answer: 12345 54321 |
| Multi-queries NIAH (MQ-NIAH) | num_queries = 2
type_key = word
type_value = number
type_haystack = essay
size_haystack ∝ context length | (essays) ......
One of the special magic numbers for long-context is: 12345.
One of the special magic numbers for large-model is: 54321.
......
What are all the special magic numbers for long-context and large-model mentioned in the provided text?
Answer: 12345 54321 |
| Variable Tracking (VT) | num_chains = 2
num_hops = 2
size_noises ∝ context length | (noises) ......
VAR X1 = 12345 ...... VAR Y1 = 54321 ......
VAR X2 = X1 ...... VAR Y2 = Y1 ......
VAR X3 = X2 ...... VAR Y3 = Y2 ......
Find all variables that are assigned the value 12345.
Answer: X1 X2 X3 |
| Common Words Extraction (CWE) | freq_cw = 2, freq_ucw = 1
num_cw = 10
num_ucw ∝ context length | aaa bbb ccc aaa ddd eee ccc fff ggg hhh iii iii ......
What are the 10 most common words in the above list?
Answer: aaa ccc iii ...... |
| Frequent Words Extraction (FWE) | α = 2
num_word ∝ context length | aaa bbb ccc aaa ddd eee ccc fff ggg aaa hhh aaa ccc iii iii ......
What are the 3 most frequently appeared words in the above coded text?
Answer: aaa ccc iii |
| Question Answering (QA) | dataset = SQuAD
num_document ∝ context length | Document 1: ...... aaa ......
Document 2: ...... bbb ......
Document 3: ...... ccc ......
Question: question
Answer: bbb |

Table 2: Task examples with flexible configurations in RULER. We use different colors to highlight queries, keys, values, and distractors in our examples.

criteria. Concretely, the retrieval capability should be (1) agnostic to the type of the "needle" and the "haystack", (2) strong enough to disregard hard distractors, and (3) of high recall when multiple items need to be retrieved. Based on these criteria, we develop four NIAH tasks. The "needle" in each of these tasks is a *key-value* pair inserted into the "haystack" (long distractor texts). The *query* is located at the end of the sequence and serves as a cue for matching the *keys* in the context and subsequently retrieving the associated *values*.

- **Single NIAH (S-NIAH):** This is the vanilla NIAH test where a single "needle"[2] needs to be retrieved from the "haystack". The *query/key/value* can take the form of words, numbers (7 digits), or UUIDs (32 digits). The "haystack" can be repeated noise sentences[3] or Paul Graham essays (Kamradt, 2023).

- **Multi-keys NIAH (MK-NIAH):** Multiple "needles" are inserted into the "haystack", and only one of them needs to be retrieved. The additional "needles" are hard distractors. The most challenging setting is a version where the "haystack" is filled with distractor needles.

---

[2]Similar to Liu et al. (2024a), we use "*the special magic number for XXX is: YYY*" as the needle due to its extendability instead of the sentence about San Francisco proposed by Kamradt (2023).

[3]Following Mohtashami & Jaggi (2023), we use "*The grass is green. The sky is blue. The sun is yellow. Here we go. There and back again.*" as noise sentences.

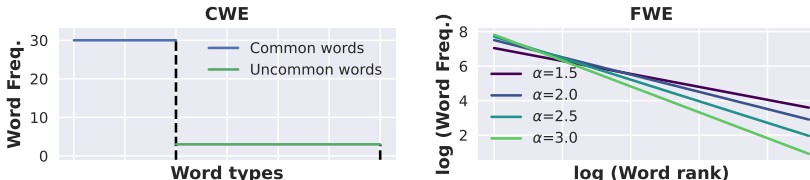

Figure 1: In aggregation tasks, we sample words from a vocabulary following the two distributions above. The common words extraction (CWE) samples from uniform distributions. In the frequent words extraction (FWE), the frequency of each word is determined by its rank in the vocabulary and the parameter $\alpha$ of Zeta distribution.

- **Multi-values NIAH (MV-NIAH):** Multiple "needles" sharing the same *key* are inserted into the "haystack". All *values* associated with the same *key* need to be retrieved.
- **Multi-queries NIAH (MQ-NIAH):** Multiple "needles" are inserted into the "haystack". All "needles" with distinct keys need to be retrieved. This is the same *multi-query associative recall* task setup used by Arora et al. (2024). Together with MV-NIAH, these two tasks evaluate the retrieval capability without missing any critical information.

### 3.2 Multi-hop Tracing: Variable Tracking (VT)

Effective discourse comprehension (van Dijk & Kintsch, 1983) is contingent upon successful recognition of newly mentioned entities and establishing the chain of references co-referring to the same entity (Karttunen, 1969) throughout the long context. We develop a new task *variable tracking* to emulate a minimal coreference chain resolution (Ng, 2010) task. This task checks the behavior of tracking relevant co-occurrence patterns and drawing skipped connections within long input. Specifically, a variable $X1$ is initialized with a value $V$, followed by a linear *chain* of variable name binding statements (e.g., $X2 = X1, X3 = X2, ...$), which are inserted at various positions of the input. The objective is to return *all* variable names pointing to the same value $V$. The task complexity can be increased by adding more hops (i.e., the times of name binding) or more chains, similar to adding hard distractors in MK-NIAH.

### 3.3 Aggregation: Common Words (CWE) and Frequent Words Extraction (FWE)

In RULER, we introduce a new category as a proxy for summarization tasks where relevant information constitutes much larger portion of the context, and the target output depends on accurate aggregation of the relevant input. Concretely, we construct an input sequence by sampling words from a pre-defined (synthetic) word list. In the common word extraction task (CWE), words are sampled from discrete uniform distributions, with the number of common words fixed while the number of uncommon words increases with the sequence length. In the frequent words extraction task (FWE), words are sampled from Zeta distribution.[4] Figure 1 shows an illustration of word frequency in the constructed input. A model needs to return the top-$K$ frequent words in the context. In CWE, $K$ equals to the number of common words. In FWE, we set $K$ to 3, as increasing $K$ leads to poor performance even at small context sizes for most models. The task complexity can be adjusted by varying the number of common words or the parameter of Zeta distribution.

### 3.4 Question Answering (QA)

The majority of existing QA datasets (Rajpurkar et al., 2018; Yang et al., 2018; Trivedi et al., 2022) are designed to answer questions based on short passages. These datasets

---

[4]We draw inspiration from Zipf's Law (Kingsley Zipf, 1932). Let $N$ be the total number of words, which is determined by the context size, the frequency of the $k$-th ranked word (the $k$-th most frequently appeared word) is $\frac{k^{-\alpha}N}{\zeta(\alpha)}$, where $\zeta(\alpha)$ is the Zeta function. We set the top-ranked word to noise.

can be extended to simulate long-context input by adding distracting information. In this task category, we insert the golden paragraphs (i.e., the paragraphs that contain answers) into paragraphs randomly sampled from the same dataset. This category is a real-world adaptation (Ivgi et al., 2023) of NIAH, where the question serves as the query, the golden paragraphs are the "needles", and the distracting paragraphs form the "haystack".

# 4    Experiments & Results

**Models & Inference setup**   We select 17 long-context LLMs, including 15 open-source models and two closed-source model (Gemini-1.5-Pro and GPT-4 ), covering diverse model sizes (7B to 8x22B with MoE architecture) and claimed context lengths (32K to 1M). Complete information about these models is included in Appendix A. We evaluate all models using vLLM (Kwon et al., 2023), an LLM serving system with efficient KV cache memory management. For all models, we run the inference in BFloat16 on 8 NVIDIA A100 GPUs with greedy decoding.

**Task configurations**   We test all models on 13 tasks ranging diverse complexities from the four categories of RULER. The test configurations have been selected (shown in Appendix B) based on a task correlational study described in Appendix C. We select these tasks as most models perform decently at short context size of 4K tokens. Our main goal is to see whether models can maintain such good performance with the scaling of context length. For each task, we evaluate each model with 500 examples generated for each length from the series (4K, 8K, 16K, 32K, 64K, 128K), while complying with each model's necessary chat template.[5] To prevent the model from refusing to answer a query or generating explanations, we append the task input with an answer prefix and check the presence of the target output with recall-based accuracy.

**Effective Context Size**   We notice large performance degradation in all models as we increase input length in RULER.  To determine the maximum context size a model can *effectively* handle, we grade each model with a fixed threshold, passing which indicates satisfactory performance at the length of evaluation. We use the performance of Llama2-7b model at the 4K context length as the threshold. We report in Table 3 the maximum length exceeding the threshold as the "effective length" along with the "claimed length".

**Model Ranking Criteria**   While the threshold-based grading reveals the discrepancy between claimed and effective length, it lacks details for fine-grained model comparisons. As such, we use a weighted average score to aggregate model performance across various context sizes. We rank models under two weighting schemes: **wAvg. (inc)** and **wAvg. (dec)** where the weight linearly increases and decreases with sequence length respectively. Ideally, the weight for each length should be determined by the length distribution of model usage, here we choose the two schemes to simulate the scenarios where longer sequences (inc) or shorter sequences (dec) dominate the distribution.

**Main Results**   We include the results of 17 long-context LMs in comparison with the Llama2-7B baseline in Table 3.[6] The performance at a certain length is the average of all 13 tasks in RULER.  The closed-source model Gemini-1.5-Pro outperforms the rest of the models by a large margin, with the effective length greater than the maximum length we have tested on. Pressure testing this model with harder version of RULER can be interesting to follow up in the future.  For the rest of the models, while they achieve nearly perfect performance on the passkey retrieval and the vanilla NIAH task (shown in Appendix E), all of them exhibit large degradation in RULER as sequence length increases and they fail to maintain performance above the Llama2-7B baseline at their claimed length. The top-ranked open-source models (Llama3.1, Qwen2 and Command-R-plus) share common configurations, such as having larger model sizes and using larger base frequencies in

---

[5]See Appendix D for model and tasks templates details.

[6]Performance of base models and breakdown by task categories can be found in Appendix F.

| Models | Claimed Length | Effective Length | 4K | 8K | 16K | 32K | 64K | 128K | Avg. | wAvg. (inc) | wAvg. (dec) |
|---|---|---|---|---|---|---|---|---|---|---|---|
| Llama2 (7B) | 4K | - | 85.6 | | | | | | | | |
| Gemini-1.5-Pro | 1M | >128K | 96.7 | 95.8 | 96.0 | 95.9 | 95.9 | 94.4 | 95.8 | 95.5(1st) | 96.1(1st) |
| GPT-4 | 128K | 64K | 96.6 | 96.3 | 95.2 | 93.2 | 87.0 | 81.2 | 91.6 | 89.0(2nd) | 94.1(2nd) |
| Llama3.1 (70B) | 128K | 64K | 96.5 | 95.8 | 95.4 | 94.8 | 88.4 | 66.6 | 89.6 | 85.5(4th) | 93.7(3rd) |
| Qwen2 (72B) | 128K | 32K | 96.9 | 96.1 | 94.9 | 94.1 | 79.8 | 53.7 | 85.9 | 79.6(9th) | 92.3(4th) |
| Command-R-plus (104B) | 128K | 32K | 95.6 | 95.2 | 94.2 | 92.0 | 84.3 | 63.1 | 87.4 | 82.7(7th) | 92.1(5th) |
| GLM4 (9B) | 1M | 64K | 94.7 | 92.8 | 92.1 | 89.9 | 86.7 | 83.1 | 89.9 | 88.0(3rd) | 91.7(6th) |
| Llama3.1 (8B) | 128K | 32K | 95.5 | 93.8 | 91.6 | 87.4 | 84.7 | 77.0 | 88.3 | 85.4(5th) | 91.3(7th) |
| GradientAI/Llama3 (70B) | 1M | 16K | 95.1 | 94.4 | 90.8 | 85.4 | 80.9 | 72.1 | 86.5 | 82.6(8th) | 90.3(8th) |
| Mixtral-8x22B (39B/141B) | 64K | 32K | 95.6 | 94.9 | 93.4 | 90.9 | 84.7 | 31.7 | 81.9 | 73.5(11th) | 90.3(9th) |
| Yi (34B) | 200K | 32K | 93.3 | 92.2 | 91.3 | 87.5 | 83.2 | 77.3 | 87.5 | 84.8(6th) | 90.1(10th) |
| Phi3-medium (14B) | 128K | 32K | 93.3 | 93.2 | 91.1 | 86.8 | 78.6 | 46.1 | 81.5 | 74.8(10th) | 88.3(11th) |
| Mistral-v0.2 (7B) | 32K | 16K | 93.6 | 91.2 | 87.2 | 75.4 | 49.0 | 13.8 | 68.4 | 55.6(13th) | 81.2(12th) |
| LWM (7B) | 1M | <4K | 82.3 | 78.4 | 73.7 | 69.1 | 68.1 | 65.0 | 72.8 | 69.9(12th) | 75.7(13th) |
| DBRX (36B/132B) | 32K | 8K | 95.1 | 93.8 | 83.6 | 63.1 | 2.4 | 0.0 | 56.3 | 38.0(14th) | 74.7(14th) |
| Together (7B) | 32K | 4K | 88.2 | 81.1 | 69.4 | 63.0 | 0.0 | 0.0 | 50.3 | 33.8(15th) | 66.7(15th) |
| LongChat (7B) | 32K | <4K | 84.7 | 79.9 | 70.8 | 59.3 | 0.0 | 0.0 | 49.1 | 33.1(16th) | 65.2(16th) |
| LongAlpaca (13B) | 32K | <4K | 60.6 | 57.0 | 56.6 | 43.6 | 0.0 | 0.0 | 36.3 | 24.7(17th) | 47.9(17th) |

Table 3: Long Context Performance (%) of selected models evaluated at length from 4K to 128K. Each score is computed by averaging accuracy of 13 tasks in RULER. The performance exceeding the Llama2-7B performance at 4K (85.6%) is underlined. The effective context length is the maximum length passing this threshold. Weighted average score (wAvg.) aggregates performance across all context sizes, with the weights linearly increasing (inc) or decreasing (dec) to simulate length distribution of real-world usage. We put the rank of each model in the subscript. More details about the selected models are in Appendix A.

RoPE (Xiong et al., 2023). Large training context window is not always necessary for good long context performance – top-ranked open-source models contain both brute-force context scaling (Llama3.1 trained on 128K context length) and inference-time length extrapolation (Qwen2 trained on 32K context length). The less performant models also include those trained on much larger context size (e.g., LWM and GradientAI/Llama3 both on 1M context length). Although LWM achieves a higher rank than Mistral-v0.2 when longer sequences receive larger weight (wAvg. inc) and shows less degradation as the context size increases, it performs worse than Llama2-7B even at 4K. This result suggests a trade-off in evaluation between absolute performance on short sequences and the relative degradation with the scaling of context size. We provide more analysis on the model size and maximum training length in section 6.

## 5 Task Error Analysis

We evaluate Yi-34B-200K with increased input lengths (up to 256K) on more complex tasks to understand the effect of task configurations and failure modes on RULER.

**Non-robustness to "needle" types.** Figure 2 (left) shows that while Yi achieves almost perfect performance when using needle of word-number pair in the standard passkey retrieval and vanilla NIAH, performance degrades when the needle takes other forms. We observe the largest degradation in the task of retrieving UUIDs, for which Yi sometimes fail to return the complete 32 digits given long (>128K) input context.

**Failure to ignore distractors.** Figure 2 (middle-left) shows that increasing the number of distracting needles steadily lowers performance, with Yi dropping by ∼40 points at 256K in the extreme version, where the context is full of irrelevant needles (#K=FULL). Error analysis reveals that Yi fails to effectively ignore the hard distractors given long input context, thus incorrectly retrieves values associated with the distractor keys. In the extreme version, Yi often returns values from the vicinity of the target, suggesting coarse match of the range but the lack of precision to locate the key when the target is in-distribution of the noises.

**Return incomplete information.** Consistent with previous works (Liu et al., 2024a; Reid et al., 2024), we notice significant degradation in performance when the model needs to

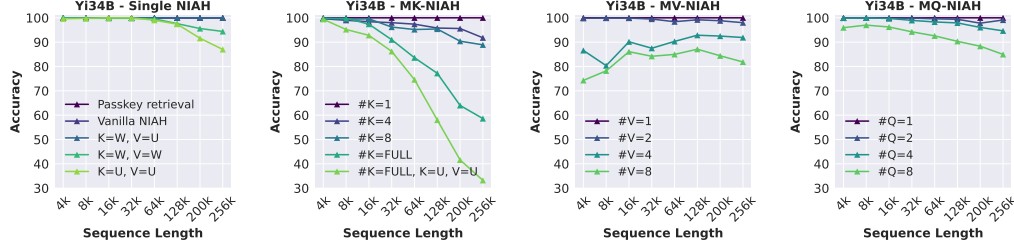

Figure 2: Performance of Yi-34B in the needle-in-a-haystack (NIAH) tasks. By default, we use word-number as the key-value pair and Paul Graham essays as the haystack. Yi is not robust to the change of needle types and degrades with the increasing amount of distractors. (W: words; N: numbers; U: UUIDs; Full: entire haystack).

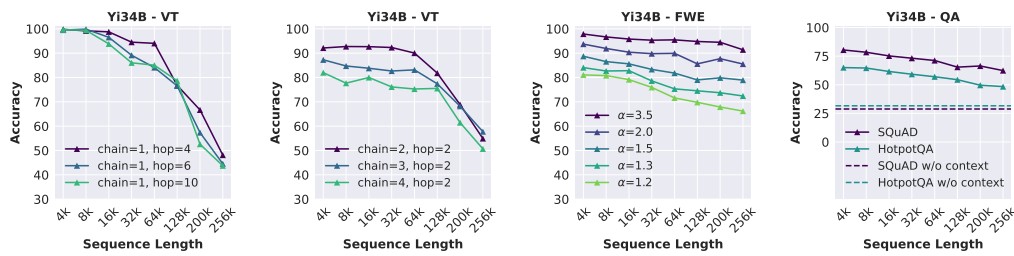

Figure 3: Performance of Yi-34B in variable tracking (VT), frequent words extraction (FWE), and QA tasks across different task complexities. Yi shows large degradation and distinct trends with scaled context size in these non-retrieval tasks, demonstrating the need to evaluate behavior beyond retrieval from context.

retrieve multiple items from a long input. For instance, increasing the number of queries from 1 to 8 drops the performance by ~15 points (Figure 2 right). When the model needs to retrieve multiple values associated with the same key (Figure 2 middle-right), Yi often outputs duplicated answers without returning the complete set of values, implying uneven associations between the key and each of its values.

**Tendency to copy from context.** We notice that Yi has a strong tendency to copy from context verbatim when scaling the input length. This tendency is most notable in *variable tracking* (VT) and *common words extraction* (CWE) where we include one in-context demonstration at the beginning of the sequence. Over 80% of Yi's output in the CWE task at 128K is simply a string copied from the one-shot example, whereas the copying is nonexistent for short sequences. [7] This copying behavior is also present in the LWM model and LongAlpaca, however it is less prevalent in other models, such as Mixtral. This finding further reinforces the need to test behaviors other than retrieval given long input context.

**Unreliable tracking within context.** For the *variable tracking* task, both adding more chains and more hops contribute to large degradation in Yi's performance. Yi consistently degrades in the more-hops setting as we increase context size (Figure 3 left), whereas the degradation in the more-chains setting is most significant for lengths greater than 128K (Figure 3 middle-left). Besides the aforementioned copying issue, Yi makes errors due to incorrectly returning empty strings or variables from other chains, implying a lack of ability to reliably trace the same entity within long context. These errors are also frequently observed in models that do not exhibit the copying behavior.

---

[7]We also experimented with removing the one-shot example. The model will simply copy the string of the beginning of the input, likely due to the attention sinks (Xiao et al., 2024b).

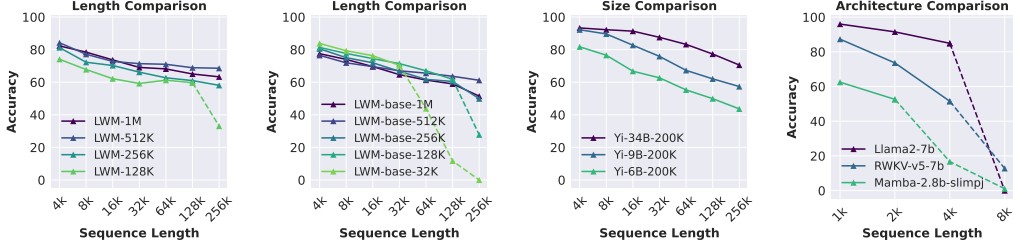

Figure 4: (**Left & middle left**): Comparison of LargeWorldModel (LWM) series trained up to various context sizes with fixed parameter size of 7B. (**Middle right**): Comparison of Yi suite models with different parameter sizes with controlled training context length of 200K. (**Right**): Performance of non-Transformer architectures lags behind the Transformer baseline Llama2-7B by large margin. Length extrapolation is presented with dashed lines.

**Failure to accurately aggregate.** We observe two common failure modes in aggregation tasks: incorrect use of parametric knowledge and inaccurate aggregation. Models that do not exhibit the copying issue in the CWE task, sometimes ignore the contextual information and instead use parametric knowledge to answer the query, especially at large context sizes. For instance, Mistral (7b-instruct-v0.2) returns high frequency words, such as "the", "an", "a", as output without counting the words in context. For the FWE task which demonstrates less the copying issue, Yi fails to correctly output the top frequent words as we decrease the $\alpha$ in Zeta distribution (Figure 3 middle-right). Decreasing $\alpha$ leads to smaller difference in frequency among words, increasing the difficulty to distinguish the top-frequent words.

**Frequent hallucination in long-context QA.** For the QA tasks, Yi's performance approaches its no-context baseline as we extend the context with distracting paragraphs (Figure 3 right). The degradation stems primarily from hallucination and reduced reliance on contextual information. We notice that, at large context sizes, model predictions sometimes are irrelevant to the question and can coincide with the answers of its no-context baseline. The overall worse performance in QA tasks confirms that the fuzzy matching between a query and a relevant paragraph in long context is a more challenging setting than the simplistic NIAH tests, where keys can be exactly located in context.

## 6 Model Analysis

**Effect of training context length.** Do models trained with larger context sizes perform better on RULER? We evaluate the suite of LargeWorldModels (Liu et al., 2024a, LWM) of equal parameter size and trained up to various context lengths. Figure 4 (left & middle-left) shows that larger context sizes overall lead to better performance, but the ranking can be inconsistent for long sequences. For instance, the model trained with 1M context size (LWM-1M) is worse than the one with 512K at length of 256K, likely due to insufficient training for adjusting to the new base frequency in RoPE. Moreover, we observe abrupt performance drops when models need to extrapolate to unseen lengths (e.g., LMW-128K given input of 256K), and almost linear degradation with input length on log scale within the max training context size.

**Effect of model size** The top models in our main results are much larger than other models. To ablate the effect of model size, we evaluate Yi-34B-200k, Yi-9B-200k, and Yi-6B-200k, all trained up to the same context length using the same data blend. Figure 4 (middle-right) shows that the 34B model is significantly better than the 6B model on RULER for both performance at length of 4K and the relative degradation, suggesting the benefit of scaling model sizes for better long-context modeling.

**Effect of architecture** We evaluate the effective context length for two models with non-Transformer architectures: RWKV-v5 (Peng et al., 2023) and Mamba-2.8B-slimpj (Gu &

Dao, 2023). We find that both models demonstrate significant degradation when extending context size to 8K, and both underperform the Transformer baseline Llama2-7B by large margins up till the length of 4K, beyond which Llama2 shows poor length extrapolation performance (Figure 4 right).

## 7   Conclusion

We present RULER, a synthetic benchmark for evaluating long-context language models. RULER contains diverse task categories, *retrieval*, *multi-hop tracing*, *aggregation* and *question answering*, providing a flexible and comprehensive evaluation of LLM's long-context capabilities. We benchmark 17 long-context LMs using RULER with context sizes ranging from 4K to 128K. Despite achieving perfect results in the widely used needle-in-a-haystack test, almost all models fail to maintain their performance in other tasks of RULER as we increase input length. We observe common failure modes at large context sizes, including the failure to ignore distractors and ineffective utilization of long context (e.g., simply copy from context or use parametric knowledge instead). We show that RULER is challenging for even the top-ranked open-source models as we increase task complexity. Our analysis further reveals the large potential for improvement on RULER and the benefit of scaling model sizes in achieving better long context capabilities.

## 8   Limitations

Despite covering more task categories than retrieval-oriented benchmarks, RULER is limited in multiple ways which we describe in detail below.

**Lack of position controlling.**   Current RULER reports a single number metric for each input length without providing the depth-level performance. The depth-level performance was evaluated by the NIAH test (Kamradt, 2023) and recent works such as LV-Eval (Yuan et al., 2024) and can be effective in revealing the lost-in-the-middle (Liu et al., 2024d) phenomenon. We are aware of this issue and plan to support the position controlling of the key information in our codebase.

**Lack of correlation with realistic long-context tasks.**   While tasks such as *variable tracking* and *frequent words extraction* were proposed to serve as proxies for real long-context natural language tasks, the lack of easy-to-evaluate realistic long-context tasks prevents us from verifying the validity of these proxies. Due to this limitation, we emphasize that RULER can be used as convenient behavioral checks of long-context language models, however it should not be preferred over more realistic settings, such as NoCHA (Karpinska et al., 2024), which also emphasize on other capabilities such as reasoning and instruction-following.

**Lack of evaluation on short context.**   In the current RULER task suite, we include tasks that most models perform reasonably well at 4k context size, and aim to observe performance degradation with the scaling of context size. This should not be misread as perfect LM capabilities at 4k context size. In fact, recent works, such as FlenQA (Levy et al., 2024), have demonstrated degrading performance when increasing their task input length to just a few thousand tokens. While increasing the task complexity in RULER leads to much worse performance at shorter context size, we did not include these results in this paper.

**Lack of verification of prompt robustness.**   Language models can be sensitive to the prompt format, however we did not extend a comprehensive study on the prompt robustness beyond preliminary testing in the early stage of this work. We also did not heavily experiment with a few fixed hyperparameters in the existing tasks, such as the length of variable names in *variable tracking* and the synthetic vocabulary size in *common word extraction* and *frequent word extraction*.

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

# A Models

We select in total 37 models for evaluation and analysis. Our results in the main text only include aligned models (GPT-4, Gemini-1.5, and 15 open-source models). Besides the aligned models, we also evaluate 7 open-source base models using RULER. We use the performance of Llama2-7b (base) and Llama2-7b (chat) at context length of 4K as the threshold for determining effective context size. In our analysis section, we evaluate in total 11 models, including model series Yi and LWM, as well as models of novel architectures, including Mamba and RWKV.

| Model | Aligned | Size | Context Length | Huggingface (Wolf et al., 2019) / API |
|---|---|---|---|---|
| GPT-4 (OpenAI: Josh Achiam et al., 2023) | ✓ | - | 128K | gpt-4-1106-preview |
| Gemini-1.5 (Reid et al., 2024) | ✓ | - | 1M | gemini-1.5-pro |
| Llama3.1 (Meta.AI, 2024b) | ✓ | 70B | 128K | meta-llama/Meta-Llama-3.1-70B-Instruct |
| Llama3.1 (Meta.AI, 2024b) | ✓ | 8B | 128K | meta-llama/Meta-Llama-3.1-8B-Instruct |
| Command-R-plus (Cohere, 2024) | ✓ | 104B | 128K | CohereForAI/c4ai-command-r-plus |
| Qwen2 (Yang et al., 2024) | ✓ | 72B | 128K | Qwen/Qwen2-72B-Instruct |
| Yi (Young et al., 2024) | ✓ | 34B | 200K | 01-ai/Yi-34B-200K |
| Mixtral-8x22B (Jiang et al., 2024) | ✓ | 39B/141B | 32K | mistralai/Mixtral-8x22B-Instruct-v0.1 |
| Mistral-v0.2 (Mistral.AI, 2023) | ✓ | 7B | 32K | mistralai/Mistral-7B-Instruct-v0.2 |
| GLM4 (GLM et al., 2024) | ✓ | 9B | 1M | THUDM/glm-4-9b-chat-1m |
| GradientAI/Llama3 (Meta.AI, 2024a) | ✓ | 70B | 1M | gradientai/Llama-3-70B-Instruct-Gradient-1048k |
| Phi3-medium (Abdin et al., 2024) | ✓ | 14B | 128K | microsoft/Phi-3-medium-128k-instruct |
| LWM (Liu et al., 2024a) | ✓ | 7B | 1M | LargeWorldModel/LWM-Text-Chat-1M |
| DBRX (Databricks, 2024) | ✓ | 36B/132B | 1M | databricks/dbrx-instruct |
| Together (Together.AI, 2023b) | ✓ | 7B | 32K | togethercomputer/Llama-2-7B-32K-Instruct |
| LongChat (Li et al., 2023a) | ✓ | 7B | 32K | lmsys/longchat-7b-v1.5-32k |
| LongAlpaca (Chen et al., 2024) | ✓ | 13B | 32K | Yukang/LongAlpaca-13B |
| Mixtral-base (Jiang et al., 2024) | ✗ | 8x7B | 32K | mistralai/Mixtral-8x7B-v0.1 |
| Mistral-base (Mistral.AI, 2023) | ✗ | 7B | 32K | alpindale/Mistral-7B-v0.2-hf |
| LWM-base (Liu et al., 2024a) | ✗ | 7B | 1M | LargeWorldModel/LWM-Text-1M |
| LongLoRA-base (Chen et al., 2024) | ✗ | 7B | 100K | Yukang/Llama-2-7b-longlora-100k-ft |
| Yarn-base(Peng et al., 2024) | ✗ | 7B | 128K | NousResearch/Yarn-Llama-2-7b-128k |
| Together-base (Together.AI, 2023a) | ✗ | 7B | 32K | togethercomputer/Llama-2-7B-32K |
| Jamba-base (AI21, 2024) | ✗ | 52B | 256K | ai21labs/Jamba-v0.1 |
| Llama2 (chat) (Touvron et al., 2023) | ✓ | 7B | 4K | meta-llama/Llama-2-7b-chat-hf |
| Llama2 (base) (Touvron et al., 2023) | ✗ | 7B | 4K | meta-llama/Llama-2-7b-hf |
| Yi series (Young et al., 2024) | ✓ | 6B,9B | 200K | 01-ai/Yi-(6B,9B)-200K |
| LWM series (Liu et al., 2024a) | ✓ | 7B | 128K,256K,512K | LargeWorldModel/LWM-Text-Chat-(128K,256K,512K) |
| LWM-base series (Liu et al., 2024a) | ✗ | 7B | 32K,128K,256K,512K | LargeWorldModel/LWM-Text-(32K,128K,256K,512K) |
| Mamba (Gu & Dao, 2023) | ✗ | 2.8B | 2K | state-spaces/mamba-2.8b-slimpj |
| RWKV (Peng et al., 2023) | ✗ | 7B | 4K | RWKV/v5-Eagle-7B-HF |

Table 4: Information of evaluated and analyzed models in RULER.

# B  Task Configurations

RULER is designed to be configurable to allow for diverse sequence lengths and task complexities. For each task, there arises combinatorially large number of configurations one can adopt. In the main text, we evaluate the models with 13 representative tasks spanning the four categories of RULER. Our task selection process is described in the next appendix section.

- **Retrieval**: In S-NIAH, we include the passkey retrieval (Mohtashami & Jaggi, 2023) and the vanilla NIAH (Kamradt, 2023), both use word-number as key-value and differ only by the background haystack. Additionally, we change the value type to UUID, for the purpose of testing model robustness at retrieving long strings from context. For MK-NIAH, we add three distractor needles into the haystack. We also include existing setups from previous works: line retrieval (Li et al., 2023a) and key-value retrieval (Liu et al., 2024d) with the haystack filled entirely with distractor needles. For MV-NIAH and MQ-NIAH, we test 4 values and queries respectively.
- **Multi-hop tracing**: For VT, we insert 1 chain with 4 name-binding hops, totally 5 variable names need to be returned.
- **Aggregation**: For CWE, in total 10 common words need to be returned, each appears 30 times whereas the uncommon words appear 3 times each. For FWE, we set $\alpha$ to 2.0 in Zeta distribution for sampling synthetic words.
- **QA**: For QA, we augment SQuAD (Rajpurkar et al., 2018) and HotpotQA (Yang et al., 2018) to simulate long-context scenario. They are representative of single-hop and multi-hop question answering tasks respectively.

| Task | Configurations | | |
|---|---|---|---|
| | **Subtask-1** | **Subtask-2** | **Subtask-3** |
| Single NIAH | type_key = word
type_value = number
type_haystack = repeat
~**passkey retrieval** | type_key = word
type_value = number
type_haystack = essay
~**vanilla NIAH** | type_key = word
type_value = uuid
type_haystack = essay |
| MK-NIAH | num_keys = 4
type_key = word
type_value = number
type_haystack = essay | num_keys = full haystack
type_key = word
type_value = number
~**line retrieval** | num_keys = full haystack
type_key = uuid
type_value = uuid
~**KV retrieval** |
| MV-NIAH | num_values = 4, type_key = word, type_value = number, type_haystack = essay | | |
| MQ-NIAH | num_queries = 4, type_key = word, type_value = number, type_haystack = essay | | |
| VT | num_chains = 1, num_hops = 4 | | |
| CWE | freq_cw = 30, freq_ucw = 3, num_cw = 10 | | |
| FWE | $\alpha$ = 2.0 | | |
| QA | dataset = SQuAD | dataset = HotpotQA | |

Table 5: Our total 13 task configurations in RULER.

# C Task Correlation Analysis

RULER is designed under the assumption that tasks across different categories are able to reveal distinct model behaviors. We conduct a preliminary correlational study to confirm the validity of task categories and guide the selection of representative tasks. We evaluate eight open-sourced models at various context sizes across 18 task configurations. Each task can then be represented with a vector of model performance at various context sizes. The 18 task vectors are then clustered via agglomorative clustering algorithm, using correlation coefficient as the distance metric. As shown in Figure 5, while certain tasks exhibit moderate correlations with others, tasks in each of the four categories (NIAH, VT, AG, QA) form cohesive clusters of their own without redundancy. We further eliminate a few tasks that correlate highly with other tasks within the same cluster, and finalize 13 tasks for later large scale evaluation.

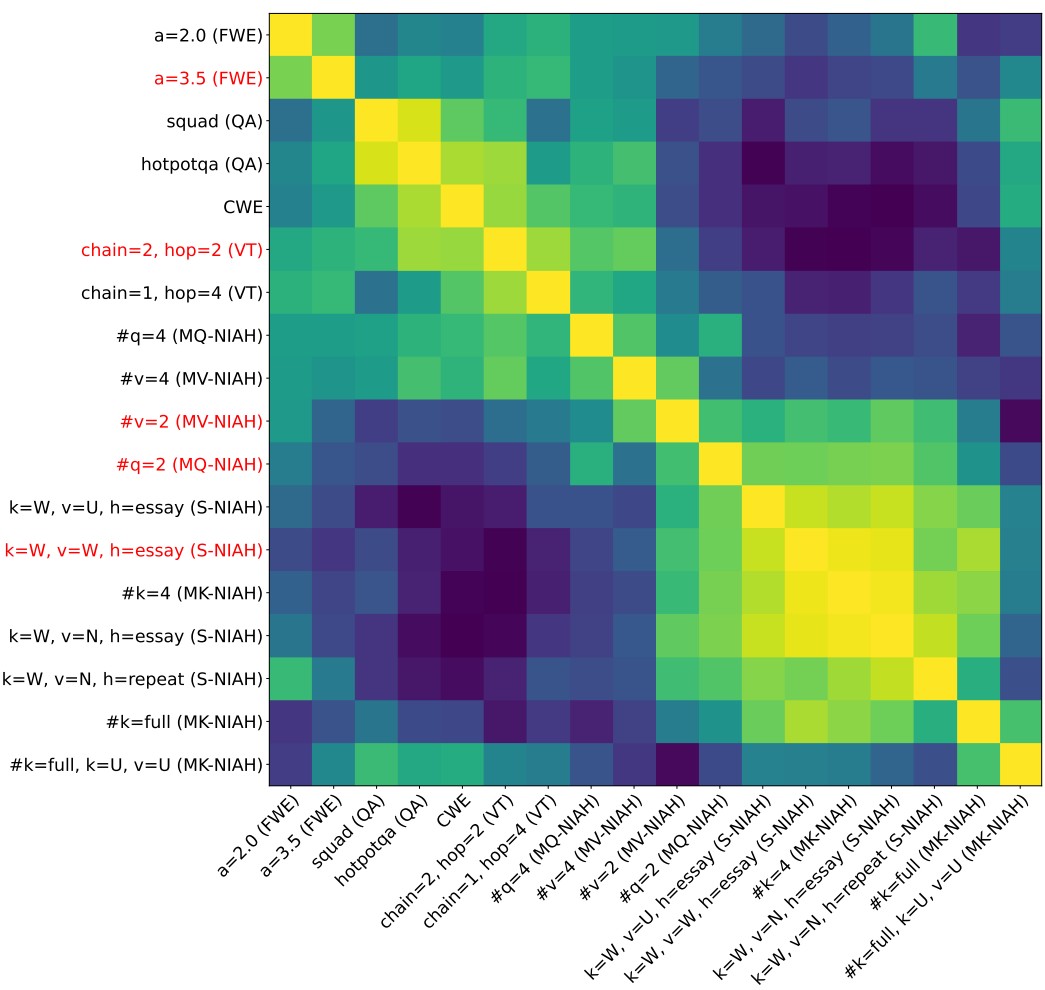

Figure 5: Correlation heatmap among 18 tasks with diverse task configurations. We remove redundant tasks (in red) and only preserve 13 representative tasks in RULER. (W: words; N: numbers; U: UUIDs; Full: entire haystack)

## D  Prompt Templates

We decompose the input prompt template into the model template in Table 6 and the task template in Table 7 8 9. The model template is the model chat format while the task template combines instruction, context, and query. To prevent models from refusing to answer our questions, we append the input with an answer prefix to elicit model responses. For VT and CWE, we use one task sample as in-context demonstration.

| Model | Template |
|---|---|
| GPT-4 | {task_template} Do not provide any explanation. Please directly give me the answer. {task_answer_prefix} |
| Yi/Base | {task_template} {task_answer_prefix} |
| Command-R | ⟨BOS_TOKEN⟩
⟨\|START_OF_TURN_TOKEN\|⟩
⟨\|USER_TOKEN\|⟩{task_template}
⟨\|END_OF_TURN_TOKEN\|⟩
⟨\|START_OF_TURN_TOKEN\|⟩
⟨\|CHATBOT_TOKEN\|⟩{task_answer_prefix} |
| LWM/LongChat | {system_prompt}  USER:  {task_template}  ASSISTANT: {task_answer_prefix} |
| GLM | [ gMASK] sop ⟨\|user\|⟩
{task_template}⟨\|assistant\|⟩
{task_answer_prefix} |
| Phi3 | ⟨\|user\|⟩
{task_template}⟨\|end\|⟩
⟨\|assistant\|⟩
{task_answer_prefix} |
| Qwen/DBRX | ⟨\|im_start\|⟩system
{system_prompt}⟨\|im_end\|⟩
⟨\|im_start\|⟩user
{task_template}⟨\|im_end\|⟩
⟨\|im_start\|⟩assistant
{task_answer_prefix} |
| Llama3/Llama3.1 | ⟨\|begin_of_text\|⟩⟨\|start_header_id\|⟩user⟨\|end_header_id\|⟩

{task_template}⟨\|eot_id\|⟩
⟨\|start_header_id\|⟩assistant⟨\|end_header_id\|⟩

{task_answer_prefix} |
| Llama2/Others | [ INST] {task_template} [ /INST] {task_answer_prefix} |

Table 6: Model chat templates. We append a task answer prefix in model response to prevent models from refusing to answer our questions. The addition of answer prefix does not break the models' chat template.

| | |
|---|---|
| S-NIAH Subtask-1 | **Task Template:**
Some special magic numbers are hidden within the following text. Make sure to memorize it. I will quiz you about the numbers afterwards.
The grass is green. The sky is blue. The sun is yellow. Here we go. There and back again. ...... One of the special magic numbers for {word} is: {number}. ......
What is the special magic number for {word} mentioned in the provided text?

**Task Answer Prefix:**
The special magic number for {word} mentioned in the provided text is |
| S-NIAH Subtask-2 | **Task Template:**
Some special magic numbers are hidden within the following text. Make sure to memorize it. I will quiz you about the numbers afterwards.
Paul Graham Essays.
...... One of the special magic numbers for {word} is: {number}. ......
What is the special magic number for {word} mentioned in the provided text?

**Task Answer Prefix:**
The special magic number for {word} mentioned in the provided text is |
| S-NIAH Subtask-3 | **Task Template:**
Some special magic words are hidden within the following text. Make sure to memorize it. I will quiz you about the words afterwards.
Paul Graham Essays.
...... One of the special magic words for {word} is: {word}. ......
What is the special magic word for {word} mentioned in the provided text?

**Task Answer Prefix:**
The special magic word for {word} mentioned in the provided text is |
| MK-NIAH Subtask-1 | **Task Template:**
Some special magic numbers are hidden within the following text. Make sure to memorize it. I will quiz you about the numbers afterwards.
Paul Graham Essays.
...... One of the special magic numbers for {word-1} is: {number-1}. ......
...... One of the special magic numbers for {word-2} is: {number-2}. ......
...... One of the special magic numbers for {word-3} is: {number-3}. ......
...... One of the special magic numbers for {word-4} is: {number-4}. ......
What is the special magic number for {word-4} mentioned in the provided text?

**Task Answer Prefix:**
The special magic number for {word-4} mentioned in the provided text is |
| MK-NIAH Subtask-2 | **Task Template:**
Some special magic numbers are hidden within the following text. Make sure to memorize it. I will quiz you about the numbers afterwards.
One of the special magic numbers for {word-1} is: {number-1}.
One of the special magic numbers for {word-2} is: {number-2}.
...... One of the special magic numbers for {word-x} is: {number-x}. ......
One of the special magic numbers for {word-n-1} is: {number-n-1}.
One of the special magic numbers for {word-n} is: {number-n}.
What is the special magic number for {word-x} mentioned in the provided text?

**Task Answer Prefix:**
The special magic number for {word-x} mentioned in the provided text is |
| MK-NIAH Subtask-3 | **Task Template:**
Some special magic uuids are hidden within the following text. Make sure to memorize it. I will quiz you about the uuids afterwards.
One of the special magic uuids for {uuid-1} is: {uuid-1}.
One of the special magic uuids for {uuid-2} is: {uuid-2}.
...... One of the special magic uuids for {uuid-x} is: {uuid-x}. ......
One of the special magic uuids for {uuid-n-1} is: {uuid-n-1}.
One of the special magic uuids for {uuid-n} is: {uuid-n}.
What is the special magic number for {uuid-x} mentioned in the provided text?

**Task Answer Prefix:**
The special magic number for {uuid-x} mentioned in the provided text is |

Table 7: S-NIAH and MK-NIAH templates.

| | |
|---|---|
| MV-NIAH | **Task Template:**
Some special magic numbers are hidden within the following text. Make sure to memorize it. I will quiz you about the numbers afterwards.
Paul Graham Essays.
...... One of the special magic numbers for {word} is: {number-1}. ......
...... One of the special magic numbers for {word} is: {number-2}. ......
...... One of the special magic numbers for {word} is: {number-3}. ......
...... One of the special magic numbers for {word} is: {number-4}. ......
What are all the special magic numbers for {word} mentioned in the provided text?

**Task Answer Prefix:**
The special magic numbers for {word} mentioned in the provided text are |
| MQ-NIAH | **Task Template:**
Some special magic numbers are hidden within the following text. Make sure to memorize it. I will quiz you about the numbers afterwards.
Paul Graham Essays.
...... One of the special magic numbers for {word-1} is: {number-1}. ......
...... One of the special magic numbers for {word-2} is: {number-2}. ......
...... One of the special magic numbers for {word-3} is: {number-3}. ......
...... One of the special magic numbers for {word-4} is: {number-4}. ......
What are all the special magic numbers for {word-1}, {word-2}, {word-3}, and {word-4} mentioned in the provided text?

**Task Answer Prefix:**
The special magic numbers for {word-1}, {word-2}, {word-3}, and {word-4} mentioned in the provided text are |
| VT | **Task Template:**
{one task example}
Memorize and track the chain(s) of variable assignment hidden in the following text.

The grass is green. The sky is blue. The sun is yellow. Here we go. There and back again.
...... VAR {X1} = {number} ......
...... VAR {X2} = {X1} ......
...... VAR {X3} = {X2} ......
...... VAR {X4} = {X3} ......
...... VAR {X5} = {X4} ......
Question: Find all variables that are assigned the value {number} in the text above.

**Task Answer Prefix:**
Answer: According to the chain(s) of variable assignment in the text above, 5 variables are assigned the value {number}, they are: |
| CWE | **Task Template:**
{one task example}
Below is a numbered list of words. In these words, some appear more often than others. Memorize the ones that appear most often.
1. word-a 2. word-b 3. word-c 4. word-a 5. word-d 6. word-a 7. word-e 8. word-f ......
Question: What are the 10 most common words in the above list?

**Task Answer Prefix:**
Answer: The top 10 words that appear most often in the list are: |
| FWE | **Task Template:**
Read the following coded text and track the frequency of each coded word. Find the three most frequently appeared coded words. ... ... word-a ... word-b ... ... ... word-c ... word-a ... word-d word-e ... word-a ... ... word-f ... ... ... ... word-g ... word-h ... word-a ... word-i ......
Question: Do not provide any explanation. Please ignore the dots '....'. What are the three most frequently appeared words in the above coded text?

**Task Answer Prefix:**
Answer: According to the coded text above, the three most frequently appeared words are: |

Table 8: MV-NIAH, MQ-NIAH, VT, CWE, and FWE templates.

| | |
|---|---|
| Single Hop QA | **Task Template:**
Answer the question based on the given documents. Only give me the answer and do not output any other words.

The following are given documents.

Document 1:
{document-1}
......
Document x:
{document-x}
......
Document n:
{document-n}

Answer the question based on the given documents. Only give me the answer and do not output any other words.

Question: question

**Task Answer Prefix:**
Answer: |
| Multi Hop QA | **Task Template:**
Answer the question based on the given documents. Only give me the answer and do not output any other words.

The following are given documents.

Document 1:
{document-1}
......
Document x:
{document-x}
......
Document y:
{document-y}
......
Document n:
{document-n}

Answer the question based on the given documents. Only give me the answer and do not output any other words.

Question: question

**Task Answer Prefix:**
Answer: |

Table 9: QA templates.

# E   Passkey Retrieval and Vanilla NIAH Results

| Models | Claimed Length | 4K | 8K | 16K | 32K | 64K | 128K | Avg. |
|---|---|---|---|---|---|---|---|---|
| Gemini-1.5 | 1M | 100.0 | 100.0 | 100.0 | 100.0 | 100.0 | 100.0 | 100.0 |
| GPT-4 | 128K | 100.0 | 100.0 | 100.0 | 100.0 | 100.0 | 100.0 | 100.0 |
| Llama3.1 (70B) | 128K | 100.0 | 100.0 | 100.0 | 100.0 | 100.0 | 97.8 | 99.6 |
| Llama3.1 (8B) | 128K | 100.0 | 100.0 | 100.0 | 100.0 | 100.0 | 100.0 | 100.0 |
| Qwen2 (72B) | 128K | 100.0 | 100.0 | 100.0 | 100.0 | 100.0 | 96.6 | 99.4 |
| Command-R-plus (104B) | 128K | 100.0 | 100.0 | 99.8 | 99.8 | 100.0 | 97.2 | 99.5 |
| GLM4 (9B) | 1M | 100.0 | 100.0 | 100.0 | 100.0 | 100.0 | 100.0 | 100.0 |
| GradientAI/Llama3 (70B) | 1M | 100.0 | 100.0 | 100.0 | 100.0 | 100.0 | 93.6 | 98.9 |
| Mixtral-8x22B (39B/141B) | 64K | 100.0 | 100.0 | 100.0 | 100.0 | 99.6 | 0.0 | 83.3 |
| Yi (34B) | 200K | 100.0 | 100.0 | 100.0 | 100.0 | 100.0 | 100.0 | 100.0 |
| Phi3-medium (14B) | 128K | 100.0 | 100.0 | 100.0 | 100.0 | 100.0 | 88.0 | 98.0 |
| Mistral-v0.2 (7B) | 32K | 100.0 | 100.0 | 100.0 | 100.0 | 99.6 | 69.6 | 94.9 |
| LWM (7B) | 1M | 100.0 | 100.0 | 100.0 | 100.0 | 100.0 | 100.0 | 100.0 |
| DBRX (36B/132B) | 32K | 100.0 | 100.0 | 100.0 | 100.0 | 0.0 | 0.0 | 66.7 |
| Together (7B) | 32K | 100.0 | 100.0 | 100.0 | 100.0 | 0.0 | 0.0 | 66.7 |
| LongChat (7B) | 32K | 100.0 | 100.0 | 100.0 | 99.4 | 0.0 | 0.0 | 66.6 |
| LongAlpaca (13B) | 32K | 88.2 | 88.6 | 86.4 | 82.4 | 0.0 | 0.0 | 57.6 |
| Mixtral-base (8x7B) | 32K | 100.0 | 100.0 | 100.0 | 100.0 | 100.0 | 46.8 | 91.1 |
| Mistral-base (7B) | 32K | 100.0 | 100.0 | 100.0 | 100.0 | 99.6 | 70.8 | 95.1 |
| Jamba-base (52B) | 256K | 100.0 | 100.0 | 100.0 | 100.0 | 100.0 | 100.0 | 100.0 |
| LWM-base (7B) | 1M | 99.8 | 100.0 | 99.6 | 99.6 | 98.2 | 96.0 | 98.9 |
| LongLoRA-base (7B) | 100K | 99.6 | 99.4 | 99.0 | 99.4 | 99.4 | 0.0 | 82.8 |
| Yarn-base (7B) | 128K | 100.0 | 100.0 | 99.0 | 100.0 | 99.2 | 39.6 | 89.6 |
| Together-base (7B) | 32K | 100.0 | 100.0 | 99.8 | 100.0 | 0.0 | 0.0 | 66.6 |

Table 10: Performance of selected aligned and base models across length 4K to 128K in passkey retrieval of RULER. Almost all models have perfect score at their claimed length.

| Models | Claimed Length | 4K | 8K | 16K | 32K | 64K | 128K | Avg. |
|---|---|---|---|---|---|---|---|---|
| Gemini-1.5 | 1M | 100.0 | 100.0 | 100.0 | 98.0 | 100.0 | 100.0 | 99.7 |
| GPT-4 | 128K | 100.0 | 100.0 | 100.0 | 100.0 | 100.0 | 100.0 | 100.0 |
| Llama3.1 (70B) | 128K | 100.0 | 100.0 | 100.0 | 100.0 | 100.0 | 99.6 | 99.9 |
| Llama3.1 (8B) | 128K | 100.0 | 100.0 | 100.0 | 100.0 | 100.0 | 99.6 | 99.9 |
| Qwen2 (72B) | 128K | 100.0 | 100.0 | 100.0 | 100.0 | 99.8 | 56.4 | 92.7 |
| Command-R-plus (35B) | 128K | 100.0 | 100.0 | 100.0 | 100.0 | 99.8 | 86.0 | 97.6 |
| GLM4 (9B) | 128K | 100.0 | 100.0 | 100.0 | 100.0 | 100.0 | 100.0 | 100.0 |
| GradientAI/Llama3 (70B) | 1M | 100.0 | 100.0 | 100.0 | 99.6 | 99.2 | 97.8 | 99.4 |
| Mixtral-8x22B (39B/141B) | 64K | 100.0 | 100.0 | 100.0 | 100.0 | 99.6 | 24.2 | 87.3 |
| Yi (34B) | 200K | 100.0 | 100.0 | 100.0 | 100.0 | 100.0 | 100.0 | 100.0 |
| Phi3-medium (14B) | 128K | 100.0 | 99.8 | 100.0 | 99.8 | 99.8 | 73.8 | 95.5 |
| Mistral-v0.2 (7B) | 32K | 100.0 | 100.0 | 100.0 | 97.0 | 70.0 | 7.4 | 79.1 |
| LWM (7B) | 1M | 100.0 | 100.0 | 100.0 | 100.0 | 100.0 | 100.0 | 100.0 |
| DBRX (36B/132B) | 32K | 100.0 | 100.0 | 90.0 | 93.2 | 0.8 | 0.0 | 64.0 |
| Together (7B) | 32K | 100.0 | 100.0 | 100.0 | 99.8 | 0.0 | 0.0 | 66.6 |
| LongChat (7B) | 32K | 100.0 | 100.0 | 97.6 | 98.4 | 0.0 | 0.0 | 66.0 |
| LongAlpaca (13B) | 32K | 90.2 | 90.2 | 88.4 | 83.4 | 0.0 | 0.0 | 58.7 |
| Mixtral-base (8x7B) | 32K | 100.0 | 100.0 | 100.0 | 100.0 | 85.2 | 34.8 | 86.7 |
| Mistral-base (7B) | 32K | 100.0 | 100.0 | 100.0 | 100.0 | 94.8 | 0.4 | 82.5 |
| Jamba-base (52B) | 256K | 100.0 | 100.0 | 98.8 | 99.8 | 99.8 | 86.4 | 97.5 |
| LWM-base (7B) | 1M | 100.0 | 99.4 | 97.8 | 98.6 | 98.2 | 98.6 | 98.8 |
| LongLoRA-base (7B) | 100K | 99.8 | 100.0 | 100.0 | 99.8 | 100.0 | 0.0 | 83.3 |
| Yarn-base (7B) | 128K | 97.4 | 97.8 | 91.4 | 85.4 | 86.6 | 20.0 | 79.8 |
| Together-base (7B) | 32K | 100.0 | 100.0 | 100.0 | 99.8 | 0.0 | 0.0 | 66.6 |

Table 11: Performance of selected aligned and base models across length 4K to 128K in vanilla NIAH of RULER. Almost all models have perfect score at their claimed length.

# F   Additional Results

| Models | Claimed Length | Effective Length | 4K | 8K | 16K | 32K | 64K | 128K | Avg. | wAvg. (inc) | wAvg. (dec) |
|---|---|---|---|---|---|---|---|---|---|---|---|
| Llama2-7B (base) | 4K | - | 79.4 | | | | | | | | |
| Mixtral-base (8x7B) | 32K | 32K | 91.8 | 91.0 | 89.5 | 85.8 | 66.9 | 29.0 | 75.7 | 66.4 (1st) | 85.0 (1st) |
| Mistral-base (7B) | 32K | 16K | 91.6 | 89.8 | 86.3 | 77.2 | 52.3 | 8.0 | 67.5 | 54.7 (4th) | 80.4 (2nd) |
| Jamba-base (52B) | 256K | 4K | 81.2 | 75.4 | 68.8 | 65.3 | 61.0 | 51.4 | 67.2 | 62.5 (3rd) | 71.8 (4th) |
| LWM-base (7B) | 1M | <4K | 77.5 | 74.0 | 69.6 | 64.6 | 61.3 | 59.0 | 67.7 | 64.4 (2nd) | 70.9 (5th) |
| LongLoRA-base (7B) | 100K | 8K | 81.9 | 80.4 | 75.6 | 65.1 | 60.8 | 0.0 | 60.6 | 49.2 (5th) | 72.0 (3rd) |
| Yarn-base (7B) | 128K | <4K | 77.3 | 67.5 | 59.0 | 47.3 | 38.6 | 13.9 | 50.6 | 40.7 (6th) | 60.5 (7th) |
| Together-base (7B) | 32K | 4K | 84.6 | 78.7 | 68.3 | 57.9 | 0.0 | 0.0 | 48.2 | 32.3 (7th) | 64.2 (6th) |

Table 12: Performance of selected base models across length 4K to 128K by averaging 13 task scores in RULER.

| Models | Claimed Length | Effective Length | 4K | 8K | 16K | 32K | 64K | 128K | Avg. | wAvg. (inc) | wAvg. (dec) |
|---|---|---|---|---|---|---|---|---|---|---|---|
| Llama2-7B (chat) | 4K | - | 96.9 | | | | | | | | |
| Gemini-1.5 | 1M | >128K | 99.8 | 99.9 | 99.6 | 99.7 | 99.7 | 99.6 | 99.7 | 99.7 (1st) | 99.7 (1st) |
| Llama3.1 (8B) | 128K | 64K | 99.9 | 99.9 | 99.8 | 99.6 | 98.7 | 92.6 | 98.4 | 97.5 (3rd) | 99.4 (2nd) |
| GLM4 (9B) | 1M | 64K | 99.4 | 99.2 | 99.5 | 99.4 | 97.3 | 94.4 | 98.2 | 97.5 (2nd) | 98.9 (3rd) |
| Llama3.1 (70B) | 128K | 64K | 100.0 | 100.0 | 99.9 | 99.6 | 98.5 | 78.9 | 96.1 | 93.5 (5th) | 98.8 (4th) |
| GPT-4 | 128K | 32K | 99.9 | 99.9 | 98.7 | 98.3 | 90.9 | 84.8 | 95.4 | 92.9 (6th) | 97.9 (5th) |
| Command-R-plus (104B) | 128K | 32K | 99.9 | 99.9 | 99.4 | 97.9 | 89.6 | 65.7 | 92.1 | 87.3 (8th) | 96.9 (6th) |
| GradientAI/Llama3 (70B) | 1M | 16K | 99.0 | 98.8 | 98.3 | 94.5 | 91.2 | 84.9 | 94.4 | 92.1 (7th) | 96.8 (7th) |
| Yi (34B) | 200K | 16K | 98.2 | 96.8 | 97.3 | 95.1 | 93.0 | 90.2 | 95.1 | 93.8 (4th) | 96.4 (8th) |
| Qwen2 (72B) | 128K | 32K | 100.0 | 99.9 | 99.9 | 99.4 | 84.5 | 48.0 | 88.6 | 81.3 (11th) | 95.9 (9th) |
| Phi3-medium (14B) | 128K | 8K | 98.7 | 98.5 | 96.6 | 95.4 | 91.9 | 51.3 | 88.7 | 82.6 (10th) | 94.9 (10th) |
| Mixtral-8x22B (39B/141B) | 64K | 16K | 99.3 | 99.1 | 97.7 | 96.7 | 89.9 | 23.8 | 84.4 | 74.8 (12th) | 94.1 (11th) |
| LWM (7B) | 1M | <4K | 92.5 | 92.1 | 87.6 | 83.7 | 84.1 | 83.4 | 87.2 | 85.5 (9th) | 89.0 (12th) |
| Mistral-v0.2 (7B) | 32K | 4K | 98.1 | 96.2 | 94.3 | 85.5 | 51.1 | 10.7 | 72.6 | 58.8 (13th) | 86.5 (13th) |
| DBRX (36B/132B) | 32K | 8K | 99.4 | 99.0 | 93.5 | 73.4 | 0.5 | 0.0 | 61.0 | 41.6 (14th) | 80.3 (14th) |
| Together (7B) | 32K | <4K | 96.2 | 89.9 | 82.3 | 80.2 | 0.0 | 0.0 | 58.1 | 40.2 (15th) | 76.0 (15th) |
| LongChat (7B) | 32K | <4K | 93.3 | 92.2 | 81.1 | 67.3 | 0.0 | 0.0 | 55.7 | 37.6 (16th) | 73.7 (16th) |
| LongAlpaca (13B) | 32K | <4K | 74.9 | 72.2 | 70.8 | 53.2 | 0.0 | 0.0 | 45.2 | 30.7 (17th) | 59.7 (17th) |
| Llama2-7B (base) | 4K | - | 90.9 | | | | | | | | |
| Mixtral-base (8x7B) | 32K | 32K | 99.9 | 99.7 | 98.4 | 94.8 | 72.1 | 29.1 | 82.3 | 71.8 (2nd) | 92.8 (1st) |
| Mistral-base (7B) | 32K | 16K | 99.3 | 97.5 | 95.7 | 89.8 | 56.8 | 10.2 | 74.9 | 61.2 (4th) | 88.6 (2nd) |
| Jamba-base (52B) | 256K | <4K | 86.4 | 80.5 | 73.7 | 72.3 | 68.1 | 56.9 | 73.0 | 68.5 (3rd) | 77.4 (5th) |
| LWM-base (7B) | 1M | <4K | 88.5 | 87.7 | 84.5 | 79.6 | 76.1 | 74.2 | 81.8 | 79.1 (1st) | 84.4 (4th) |
| LongLoRA-base (7B) | 100K | 16K | 95.3 | 95.6 | 92.7 | 81.5 | 76.2 | 0.0 | 73.5 | 60.6 (5th) | 86.5 (3rd) |
| Yarn-base (7B) | 128K | <4K | 89.9 | 86.1 | 78.4 | 59.0 | 49.5 | 17.5 | 63.4 | 51.7 (6th) | 75.1 (7th) |
| Together-base (7B) | 32K | 8K | 95.4 | 91.5 | 86.1 | 75.1 | 0.0 | 0.0 | 58.0 | 39.9 (7th) | 76.2 (6th) |

Table 13: Performance of selected aligned and base models across length 4K to 128K by averaging 8 task scores in Retrieval (NIAH) of RULER.

| Models | Claimed Length | Effective Length | 4K | 8K | 16K | 32K | 64K | 128K | Avg. | wAvg. (inc) | wAvg. (dec) |
|---|---|---|---|---|---|---|---|---|---|---|---|
| Llama2-7B (chat) | 4K | - | 89.7 | | | | | | | | |
| GPT-4 | 128K | 128K | 100.0 | 100.0 | 100.0 | 100.0 | 100.0 | 99.6 | 99.9 | 99.9$_{(2nd)}$ | 100.0$_{(1st)}$ |
| Gemini-1.5 | 1M | >128K | 100.0 | 100.0 | 100.0 | 100.0 | 99.6 | 100.0 | 99.9 | 99.9$_{(1st)}$ | 100.0$_{(2nd)}$ |
| Command-R-plus (104B) | 128K | 128K | 100.0 | 100.0 | 100.0 | 100.0 | 99.9 | 97.2 | 99.5 | 99.2$_{(3rd)}$ | 99.8$_{(3rd)}$ |
| GLM4 (9B) | 1M | >128K | 99.9 | 99.6 | 99.8 | 99.8 | 99.6 | 97.7 | 99.4 | 99.1$_{(4th)}$ | 99.7$_{(4th)}$ |
| Qwen2 (72B) | 128K | 64K | 100.0 | 100.0 | 100.0 | 100.0 | 95.2 | 79.0 | 95.7 | 92.9$_{(5th)}$ | 98.5$_{(5th)}$ |
| Llama3.1 (70B) | 128K | 64K | 100.0 | 100.0 | 100.0 | 100.0 | 99.9 | 59.2 | 93.2 | 88.3$_{(8th)}$ | 98.0$_{(6th)}$ |
| Llama3.1 (8B) | 128K | 64K | 99.9 | 99.7 | 99.7 | 98.8 | 97.6 | 70.4 | 94.4 | 90.7$_{(6th)}$ | 98.0$_{(7th)}$ |
| GradientAI/Llama3 (70B) | 1M | 64K | 100.0 | 100.0 | 100.0 | 100.0 | 99.7 | 56.2 | 92.6 | 87.4$_{(9th)}$ | 97.9$_{(8th)}$ |
| Yi (34B) | 200K | 64K | 99.8 | 99.2 | 98.8 | 94.5 | 92.5 | 76.8 | 93.6 | 90.3$_{(7th)}$ | 96.9$_{(9th)}$ |
| Mixtral-8x22B (39B/141B) | 64K | 64K | 100.0 | 100.0 | 99.8 | 98.6 | 96.4 | 0.0 | 82.5 | 70.3$_{(10th)}$ | 94.7$_{(10th)}$ |
| Phi3-medium (14B) | 128K | 16K | 99.6 | 99.2 | 98.4 | 82.1 | 53.6 | 26.0 | 76.5 | 64.1$_{(11th)}$ | 88.9$_{(11th)}$ |
| Mistral-v0.2 (7B) | 32K | 16K | 98.9 | 96.0 | 92.2 | 85.0 | 74.5 | 0.0 | 74.4 | 60.9$_{(12th)}$ | 87.9$_{(12th)}$ |
| LongChat (7B) | 32K | 8K | 97.6 | 93.5 | 83.4 | 62.4 | 0.0 | 0.0 | 56.2 | 37.4$_{(14th)}$ | 75.0$_{(13th)}$ |
| DBRX (36B/132B) | 32K | 8K | 100.0 | 99.0 | 72.5 | 45.8 | 0.0 | 0.0 | 52.9 | 33.3$_{(15th)}$ | 72.5$_{(14th)}$ |
| LWM (7B) | 1M | <4K | 84.4 | 80.1 | 67.2 | 52.2 | 45.9 | 15.2 | 57.5 | 46.5$_{(13th)}$ | 68.6$_{(15th)}$ |
| Together (7B) | 32K | <4K | 89.2 | 88.8 | 48.3 | 16.6 | 0.0 | 0.0 | 40.5 | 22.8$_{(16th)}$ | 58.2$_{(16th)}$ |
| LongAlpaca (13B) | 32K | <4K | 8.5 | 2.1 | 18.2 | 17.0 | 0.0 | 0.0 | 7.6 | 6.5$_{(17th)}$ | 8.8$_{(17th)}$ |
| Llama2-7B (base) | 4K | - | 58.8 | | | | | | | | |
| Mixtral-base (8x7B) | 32K | 64K | 100.0 | 99.9 | 100.0 | 98.4 | 87.3 | 43.3 | 88.1 | 80.5$_{(2nd)}$ | 95.8$_{(1st)}$ |
| Mistral-base (7B) | 32K | 64K | 99.0 | 98.4 | 96.5 | 89.1 | 86.1 | 0.0 | 78.2 | 65.4$_{(4th)}$ | 91.0$_{(2nd)}$ |
| Jamba-base (52B) | 256K | 128K | 87.5 | 87.6 | 86.2 | 88.1 | 86.0 | 77.8 | 85.5 | 84.3$_{(1st)}$ | 86.7$_{(3rd)}$ |
| LWM-base (7B) | 1M | 128K | 80.2 | 82.7 | 79.3 | 76.4 | 70.7 | 66.1 | 75.9 | 73.3$_{(3rd)}$ | 78.5$_{(4th)}$ |
| LongLoRA-base (7B) | 100K | 64K | 92.5 | 87.4 | 73.1 | 56.0 | 69.2 | 0.0 | 63.0 | 50.3$_{(5th)}$ | 75.8$_{(5th)}$ |
| Yarn-base (7B) | 128K | 4K | 84.6 | 43.6 | 24.8 | 43.0 | 20.9 | 0.0 | 36.1 | 24.9$_{(7th)}$ | 47.4$_{(7th)}$ |
| Together-base (7B) | 32K | 16K | 95.0 | 90.6 | 69.6 | 43.2 | 0.0 | 0.0 | 49.7 | 31.3$_{(6th)}$ | 68.1$_{(6th)}$ |

Table 14: Performance of selected aligned and base models across length 4K to 128K in Multi-hop tracing (VT) of RULER.

| Models | Claimed Length | Effective Length | 4K | 8K | 16K | 32K | 64K | 128K | Avg. | wAvg. (inc) | wAvg. (dec) |
|---|---|---|---|---|---|---|---|---|---|---|---|
| Llama2-7B-chat | 4K | - | 84.8 | | | | | | | | |
| Gemini-1.5 | 1M | >128K | 97.7 | 97.7 | 97.6 | 98.6 | 97.3 | 90.9 | 96.6 | 95.8$_{(1st)}$ | 97.4$_{(1st)}$ |
| GPT-4 | 128K | 64K | 99.0 | 98.3 | 98.0 | 95.0 | 90.1 | 79.7 | 93.4 | 90.4$_{(2nd)}$ | 96.3$_{(2nd)}$ |
| Qwen2 (72B) | 128K | 32K | 99.3 | 98.0 | 93.1 | 97.4 | 78.5 | 70.3 | 89.4 | 84.7$_{(3rd)}$ | 94.2$_{(3rd)}$ |
| Mixtral-8x22B (39B/141B) | 64K | 32K | 97.8 | 96.9 | 94.8 | 88.2 | 83.3 | 69.7 | 88.5 | 84.0$_{(4th)}$ | 92.9$_{(4th)}$ |
| Command-R-plus (104B) | 128K | 32K | 98.2 | 96.9 | 95.2 | 90.3 | 82.5 | 59.5 | 87.1 | 81.3$_{(5th)}$ | 92.8$_{(5th)}$ |
| Llama3.1 (70B) | 128K | 32K | 99.9 | 98.3 | 98.4 | 97.1 | 66.3 | 39.8 | 83.3 | 73.8$_{(6th)}$ | 92.8$_{(6th)}$ |
| Phi3-medium (14B) | 128K | 16K | 90.8 | 95.1 | 90.3 | 82.4 | 62.1 | 43.8 | 77.4 | 69.3$_{(7th)}$ | 85.6$_{(7th)}$ |
| Yi (34B) | 200K | 16K | 91.4 | 90.9 | 86.2 | 75.3 | 58.5 | 43.4 | 74.3 | 66.0$_{(8th)}$ | 82.6$_{(8th)}$ |
| GLM4 (9B) | 1M | 8K | 93.5 | 85.2 | 78.5 | 68.1 | 58.3 | 49.7 | 72.2 | 64.8$_{(9th)}$ | 79.6$_{(9th)}$ |
| GradientAI/Llama3 (70B) | 1M | 8K | 96.4 | 94.7 | 74.9 | 57.0 | 45.1 | 41.4 | 68.3 | 57.7$_{(10th)}$ | 78.8$_{(10th)}$ |
| Llama3.1 (8B) | 128K | 8K | 97.0 | 90.1 | 79.2 | 54.1 | 43.5 | 36.2 | 66.7 | 55.5$_{(11th)}$ | 77.8$_{(11th)}$ |
| Mistral-v0.2 (7B) | 32K | 8K | 94.3 | 90.4 | 77.4 | 48.5 | 42.4 | 33.7 | 64.4 | 53.1$_{(12th)}$ | 75.8$_{(12th)}$ |
| DBRX (36B/132B) | 32K | 8K | 94.5 | 94.7 | 73.7 | 48.7 | 4.1 | 0.0 | 52.6 | 34.3$_{(13th)}$ | 70.9$_{(13th)}$ |
| Together (7B) | 32K | <4K | 82.3 | 64.5 | 43.3 | 34.8 | 0.0 | 0.0 | 37.5 | 22.9$_{(16th)}$ | 52.1$_{(14th)}$ |
| LongChat (7B) | 32K | <4K | 74.3 | 50.7 | 46.7 | 51.1 | 0.0 | 0.0 | 37.1 | 24.8$_{(15th)}$ | 49.5$_{(15th)}$ |
| LWM (7B) | 1M | <4K | 61.3 | 43.6 | 38.3 | 32.8 | 29.1 | 29.1 | 39.0 | 34.0$_{(14th)}$ | 44.0$_{(16th)}$ |
| LongAlpaca (13B) | 32K | <4K | 33.0 | 27.0 | 26.0 | 23.2 | 0.0 | 0.0 | 18.2 | 12.3$_{(17th)}$ | 24.1$_{(17th)}$ |
| Llama2-7B (base) | 4K | - | 73.1 | | | | | | | | |
| Mixtral-base (8x7B) | 32K | 32K | 96.5 | 94.8 | 93.1 | 87.8 | 68.6 | 24.3 | 77.5 | 66.9$_{(1st)}$ | 88.1$_{(1st)}$ |
| Mistral-base (7B) | 32K | 16K | 94.8 | 93.1 | 81.6 | 53.3 | 36.7 | 9.2 | 61.4 | 46.5$_{(2nd)}$ | 76.3$_{(2nd)}$ |
| Jamba-base (52B) | 256K | 4K | 75.9 | 63.5 | 51.7 | 38.5 | 33.3 | 28.0 | 48.5 | 40.3$_{(3rd)}$ | 56.6$_{(3rd)}$ |
| LWM-base (7B) | 1M | <4K | 67.1 | 48.4 | 36.0 | 26.3 | 21.5 | 18.7 | 36.3 | 28.4$_{(5th)}$ | 44.2$_{(5th)}$ |
| LongLoRA-base (7B) | 100K | <4K | 70.3 | 64.4 | 50.7 | 39.9 | 29.4 | 0.0 | 42.4 | 31.3$_{(4th)}$ | 53.6$_{(4th)}$ |
| Yarn-base (7B) | 128K | <4K | 70.6 | 49.2 | 28.9 | 20.5 | 17.0 | 2.1 | 31.4 | 20.7$_{(6th)}$ | 42.0$_{(6th)}$ |
| Together-base (7B) | 32K | <4K | 69.1 | 53.0 | 19.9 | 20.6 | 0.0 | 0.0 | 27.1 | 15.1$_{(7th)}$ | 39.1$_{(7th)}$ |

Table 15: Performance of selected aligned and base models across length 4K to 128K by averaging 2 task scores in Aggregation (CWE/FWE) of RULER.

| Models | Claimed Length | Effective Length | 4K | 8K | 16K | 32K | 64K | 128K | Avg. | wAvg. (inc) | wAvg. (dec) |
|---|---|---|---|---|---|---|---|---|---|---|---|
| Llama2-7B (chat) | 4K | - | 49.7 | | | | | | | | |
| Gemini-1.5 | 1M | >128K | 81.9 | 75.9 | 77.8 | 75.9 | 77.6 | 74.1 | 77.2 | 76.3$_{(1st)}$ | 78.0$_{(1st)}$ |
| GPT-4 | 128K | 128K | 79.0 | 78.0 | 76.0 | 68.0 | 61.6 | 59.0 | 70.3 | 66.5$_{(4th)}$ | 74.0$_{(2nd)}$ |
| Qwen2 (72B) | 128K | 64K | 80.8 | 76.9 | 74.1 | 66.9 | 54.5 | 47.2 | 66.7 | 61.0$_{(8th)}$ | 72.5$_{(3rd)}$ |
| GradientAI/Llama3 (70B) | 1M | >128K | 75.6 | 73.9 | 72.4 | 69.9 | 66.0 | 59.8 | 69.6 | 67.1$_{(3rd)}$ | 72.1$_{(4th)}$ |
| Llama3.1 (70B) | 128K | 64K | 77.2 | 74.8 | 72.3 | 70.4 | 64.2 | 47.6 | 67.8 | 63.4$_{(7th)}$ | 72.1$_{(5th)}$ |
| GLM4 (9B) | 1M | >128K | 74.7 | 71.3 | 71.9 | 68.5 | 66.3 | 63.6 | 69.4 | 67.6$_{(2nd)}$ | 71.1$_{(6th)}$ |
| Mixtral-8x22B (39B/141B) | 64K | 64K | 76.6 | 73.5 | 71.8 | 66.5 | 59.7 | 40.8 | 64.8 | 59.4$_{(9th)}$ | 70.2$_{(7th)}$ |
| Yi (34B) | 200K | >128K | 72.7 | 71.5 | 68.4 | 66.2 | 64.1 | 59.9 | 67.1 | 65.0$_{(5th)}$ | 69.2$_{(8th)}$ |
| Llama3.1 (8B) | 128K | 128K | 74.1 | 70.1 | 67.3 | 65.8 | 63.7 | 58.8 | 66.6 | 64.3$_{(6th)}$ | 68.9$_{(9th)}$ |
| Command-R-plus (104B) | 128K | 64K | 73.4 | 72.3 | 69.4 | 65.9 | 57.0 | 39.2 | 62.9 | 57.6$_{(10th)}$ | 68.1$_{(10th)}$ |
| Phi3-medium (14B) | 128K | 64K | 70.9 | 67.2 | 66.1 | 59.3 | 54.2 | 38.0 | 59.3 | 54.3$_{(12th)}$ | 64.3$_{(11th)}$ |
| Mistral-v0.2 (7B) | 32K | 32K | 72.4 | 70.0 | 65.7 | 57.6 | 34.4 | 13.3 | 52.2 | 42.5$_{(13th)}$ | 62.0$_{(12th)}$ |
| LWM (7B) | 1M | >128K | 61.2 | 57.8 | 56.7 | 55.4 | 54.7 | 52.6 | 56.4 | 55.1$_{(11th)}$ | 57.7$_{(13th)}$ |
| DBRX (36B/132B) | 32K | 16K | 76.0 | 69.4 | 59.4 | 45.0 | 9.6 | 0.0 | 43.2 | 29.6$_{(14th)}$ | 56.9$_{(14th)}$ |
| Together (7B) | 32K | 16K | 61.1 | 58.3 | 54.2 | 45.6 | 0.0 | 0.0 | 36.5 | 24.9$_{(15th)}$ | 48.2$_{(15th)}$ |
| LongAlpaca (13B) | 32K | 16K | 57.2 | 53.5 | 49.7 | 39.0 | 0.0 | 0.0 | 33.2 | 22.3$_{(16th)}$ | 44.1$_{(16th)}$ |
| LongChat (7B) | 32K | 8K | 54.5 | 53.6 | 47.6 | 34.0 | 0.0 | 0.0 | 31.6 | 21.0$_{(17th)}$ | 42.3$_{(17th)}$ |
| Llama2-7B (base) | 4K | - | 48.6 | | | | | | | | |
| Mixtral-base (8x7B) | 32K | 4K | 50.8 | 47.7 | 45.3 | 41.3 | 34.4 | 26.4 | 41.0 | 37.0$_{(3rd)}$ | 44.9$_{(3rd)}$ |
| Mistral-base (7B) | 32K | 8K | 53.5 | 51.0 | 48.4 | 44.7 | 32.8 | 2.2 | 38.8 | 31.3$_{(4th)}$ | 46.3$_{(2nd)}$ |
| Jamba-base (52B) | 256K | 32K | 62.7 | 60.6 | 57.9 | 52.6 | 47.5 | 39.6 | 53.5 | 49.7$_{(1st)}$ | 57.3$_{(1st)}$ |
| LWM-base (7B) | 1M | <4K | 42.7 | 40.2 | 38.7 | 37.1 | 37.3 | 34.6 | 38.4 | 37.2$_{(2nd)}$ | 39.6$_{(4th)}$ |
| LongLoRA-base (7B) | 100K | <4K | 34.5 | 32.1 | 33.6 | 29.4 | 26.1 | 0.0 | 26.0 | 21.3$_{(6th)}$ | 30.6$_{(6th)}$ |
| Yarn-base (7B) | 128K | <4K | 29.7 | 23.5 | 28.6 | 29.7 | 25.5 | 18.1 | 25.9 | 24.6$_{(5th)}$ | 27.1$_{(7th)}$ |
| Together-base (7B) | 32K | 4K | 52.0 | 47.5 | 44.6 | 33.6 | 0.0 | 0.0 | 29.6 | 19.8$_{(7th)}$ | 39.5$_{(5th)}$ |

Table 16: Performance of selected aligned and base models across length 4K to 128K by averaging 2 task scores in Question Answering of RULER.

