# OpenReview forum: "RULER: What’s the Real Context Size of Your Long-Context Language Models?"
_colmweb.org/COLM/2024/Conference — COLM_

### Official Review · Reviewer_4oYN · 2024-04-12

**Rating:** 7
**Confidence:** 3
**Ethics Flag:** 1

**Summary:**

This paper provides a new synthetic long-context LLM benchmark dubbed RULER. Compared to previous benchmarks, diverse newly designed tasks have been applied with flexible context lengths, characterized by different difficulty levels. Extensive experiments have been conducted on several long-context LLMs under different task settings.

**Questions To Authors:**

1. How can we make sure the synthetic datasets do not have more preference for certain LLMs, regardless of their long-context learning ability?
2. How to define the complexity of a task? It seems to be more complicated than simply defining it as the length of the context, some other factors might contribute such as distraction level. How to control this factor?
3. For the aggregation task, does LLM on different vocabularies have different performances?

**Reasons To Accept:**

1.  A new benchmark with more difficult tasks has been provided. The benchmark could better distinguish the long-context capability of LLMs compared to existing benchmarks,  which might contribute to future long-context research.
2. Extensive experiments have been conducted, with deep analysis with demonstration of RULER's capability.
3. Insightful findings on the long-context ability of LLMs might be helpful for future researchers.

**Reasons To Reject:**

1. The authors could strictly control variables while synthesizing the dataset. For example, the task of Retrieval, where keys and values are put in the context might be also influential to the performance of LLMs.
2. The synthetic datasets for one task all have the same template, which might not be diverse enough.

---

> ### Author Rebuttal · Authors · 2024-05-30
>
> We thank reviewer 4oYN for their thoughtful comments.
>
> The reviewer raises concern on the template used for all models. We employ the same template to enforce a controlled evaluation setup for all models. While the diversity of the prompts can be a valid concern for evaluating large language models, the robustness to various prompts is an orthogonal research direction which we believe to be out-of-scope of our paper. However, we agree with the reviewer that evaluating models on additional prompts can further strengthen our results.
>
> To answer the questions raised by the reviewer:
> 1. In the current version of RULER, we purposefully select the synthetic tasks on which the majority of the models perform decently with short input length (<4k) so that the major differences we observe are due to the scaling of context size. We did not include harder tasks in RULER (e.g., those pressure testing Yi-34B in the analysis section) as those tasks will unfairly put some models in disadvantage even at short sequences.
> 2. As discussed in section 5, the reviewer is correct that the number of distractors indeed also impacts the complexity of a task. Within a constrained domain as in RULER, we believe the task complexity is a function of at least two factors: the number of target output tokens and the signal-to-noise ratio in the context. The number of target output tokens can be changed by changing the number of returned needles, the number of hops in variable tracking, and the number of top-frequent words to be returned in CWE/FWE. The signal-to-noise ratio can be controlled by the number of distractors in NIAH and variable tracking, the frequency of common words in the CWE task and the hyperparameter alpha in the FWE task. We plan to add more detailed relevant discussion in the next version.
> 3. In our preliminary experiments, we observe very small variances evaluating the same model on different synthetic vocabularies since we fix the size of the synthetic vocabulary and the length of each synthetic word. While these two hyperparameters can also be varied, we keep them fixed in our experiments to have a controlled setup. It can be interesting to investigate how these two hyperparameters impact model performance. We thank the reviewer for bringing this up and will add relevant discussion in our revised version.

---

> ### Comment · Reviewer_4oYN · 2024-06-03
>
> Thanks for your considerate response, most of my concerns have been addressed. I would like to see what the additional discussion will provide us in the future version.

---

### Official Review · Reviewer_HSfg · 2024-04-23

**Rating:** 7
**Confidence:** 4
**Ethics Flag:** 1

**Summary:**

The manuscript introduces RULER, a comprehensive synthetic benchmark designed to evaluate the real-world capabilities of long-context language models (LLMs). The authors argue that traditional benchmarks do not adequately measure true contextual understanding, especially across longer text spans. To address this gap, RULER integrates complex tasks including multi-hop tracing and information aggregation, aiming to provide a more accurate assessment of LLMs' long-context processing abilities.

**Reasons To Accept:**

1) The evaluation is thorough, testing ten different LLMs across 13 tasks that vary in complexity and length. This comprehensive testing provides a clear picture of how well these models perform under rigorous and varied conditions.
2) The manuscript offers valuable insights into the performance degradation of LLMs as context size increases. These insights are crucial for understanding the limitations of current LLMs and could guide future research and development in the field.

**Reasons To Reject:**

1) The dataset constructed in this article primarily extends the range of tasks and enriches the current evaluation dataset. However, it shows limited innovation in terms of novel methodologies or insights into language model capabilities.
2) While RULER's synthetic tasks provide controlled conditions for testing, the absence of a comparative analysis with benchmarks that use realistic data might limit understanding of how these findings translate to real-world applications.
3) It is recommended to supplement the analysis with details on the usage of graphics memory and inference speed for different contexts and models to provide a more comprehensive evaluation of model efficiency.

---

> ### Author Rebuttal · Authors · 2024-05-30
>
> We thank reviewer HSfg for their thoughtful comments on our submission.
>
> The reviewer mentions the absence of a comparative study between RULER and real-world applications. We would like to mention that real-world long-context tasks usually have fixed input lengths, thus it can be difficult to perform evaluation like we do with RULER, where we evaluate models with increased context size while maintaining the same and known upper-bound performance. Moveover, existing long-context tasks employ imperfect automatic metrics (e.g. ROUGE for book-level summarization), whereas gold human evaluation for long-context tasks can be expensive and the topic itself warrants a separate research work 1]. We agree with the reviewer that the correlation between performance on RULER and other real-world applications can be fruitful to explore. However, due to the scarcity of high-quality naturalistic long-context tasks that are easy to evaluate, we decide that this particular direction is out-of-scope of our paper and can be a promising direction to pursue in the future.
>
> We agree with the reviewer that additionally profiling inference speed and memory footprint can give valuable information about performance-efficiency trade-off. We will add the support of comparing inference efficiency in our open-source code repository in the future.
>
> Reference:
> [1] BooookScore: A systematic exploration of book-length summarization in the era of LLMs

---

### Official Review · Reviewer_GXoR · 2024-04-30

**Rating:** 7
**Confidence:** 4
**Ethics Flag:** 1

**Summary:**

This paper presents RULER, a new benchmark to comprehensively evaluate the capabilities of long-context language models through synthetic tasks with flexible configurations. These tasks expand the needle-in-a-haystack test which is currently popular. Empirical results reveal that current long-text models exhibit large degradation on these complex tasks, and training on longer sequences does not always lead to better performance.

**Reasons To Accept:**

- Providing a novel benchmark that incorporates diverse tasks to test long-context abilities of large language models.
- Thorough evaluation and error analysis of a range of models, providing a comprehensive understanding of their performance characteristics.
- Well-written paper with clear motivation, illustration of tasks, and presentation of results.

**Reasons To Reject:**

- There are many similar benchmarks, and the significance of this paper is not prominent. Further discussion may be needed on the advantages of this paper's benchmark and why these advantages are important.
- The synthetic nature of tasks may not fully capture the challenges of real-world long-context applications. It may sound easy to construct more complex synthesis tasks, but we need more insights about real long-context requirements. For example, How well do the synthetic tasks in RULER correlate with performance on real-world long-context applications? Is it necessary for long-context models to complete these complex tasks such as extreme long chain of binding statements?

---

> ### Author Rebuttal · Authors · 2024-05-30
>
> We thank reviewer GXoR for their encouraging comments and thoughtful advice.
>
> We agree with the reviewer that there are similar long-context benchmarks and thus have included a brief comparison in section 1 Table 1.
> - Compared to existing synthetic evaluation such as passkey retrieval and needle-in-a-haystack, RULER covers diverse task categories such as tracing and aggregation, testing LM capabilities beyond simple in-context recall.
> - Compared to existing realistic long-context benchmarks, RULER reduces the impact of parametric knowledge by maximizing the reliance on the contextual information. Evaluating on input that has large overlap with the pre-training data (e.g., book data used by other benchmarks) entangles context utilization with memorized parametric knowledge, whereas testing on fully synthetic data allows us to test a model’s behavior while minimizing the memorization effect.
> - Compared to existing realistic long-context benchmarks, RULER also has the key advantage of allowing for flexible context length and varied task complexity. Existing benchmarks mostly adopt fixed input length, which can be too short to test “long”-context language models. In contrast, input can be adjusted to arbitrary length in RULER. Moreover, the number of distractors and the amount of relevant information can also be controlled by task hyperparameters, which lead to different task complexities.
>
> Reviewer mentions a potential gap between the synthetic nature of RULER and the intricacy of real-world long-context applications. We acknowledge the limitation of synthetic tasks and would like to emphasize in the revised version that RULER can serve as diagnostic tests of LM capabilities (e.g., retrieval, tracing, and aggregation), however it cannot replace real-world long-context tasks. We agree with the reviewer that it can be interesting to see the correlation between RULER and other real-world long-context tasks, since many tasks in RULER are heavily motivated by realistic natural language tasks (e.g., variable tracking motivated by coreference chain resolution, and frequent word extraction motivated by summarization) after all. Due to the scarcity of high-quality naturalistic long-context tasks that are easy to evaluate, we decide that this particular direction is out-of-scope of our paper and can be a promising direction to pursue in the future.
>
> We thank the reviewer for bringing these two points up and will add more detailed discussions in our revised version.

---

> > ### Comment · Reviewer_GXoR · 2024-06-05
> >
> > Thank you very much for your responses. I will keep my current rating score and tend to accept the paper.

---

### Official Review · Reviewer_cgKY · 2024-05-10

**Rating:** 7
**Confidence:** 4
**Ethics Flag:** 1

**Summary:**

The paper introduces a new synthetic benchmark, RULER, designed to evaluate the long-context modeling capabilities of language models (LMs). The authors argue that existing tests, such as the needle-in-a-haystack (NIAH), are insufficient to comprehensively assess LMs' understanding of long contexts. RULER expands beyond simple retrieval tasks to include multi-hop tracing, aggregation, and question answering, allowing for a more nuanced evaluation of model performance across various context lengths and complexities. The study involves testing ten long-context LMs with 13 representative tasks and finds that despite near-perfect accuracy in NIAH, all models experience significant performance drops as context length increases. The paper's analysis reveals that only a few models can maintain satisfactory performance at 32K tokens, and there is substantial room for improvement, especially as input length and task complexity increase.

**Questions To Authors:**

na

**Reasons To Accept:**

1. RULER provides a more thorough evaluation of LMs by including diverse tasks that go beyond simple retrieval, offering a better understanding of how models handle long contexts. The benchmark allows for adjustments in sequence length and task complexity, making it adaptable for various research needs and future model evaluations.

2. The paper presents detailed results and analysis across different models and context sizes, giving insights into the limitations and performance degradation of current LMs.

3. The study reveals that most models struggle with complex tasks in long contexts, despite their claims of handling large context sizes, which is valuable information for both researchers and practitioners.

**Reasons To Reject:**

1. The study focuses on a selection of models, which might not represent the entire spectrum of available LMs, possibly missing insights from a broader set. Actually, there are lots of models that specially optimized for long context scenarios, such as Claude and Moonshot, which the performance needs to be discussed.

---

> ### Author Rebuttal · Authors · 2024-05-30
>
> We thank reviewer cgKY for their encouraging and thoughtful comments.
>
> We agree with the reviewer that evaluating the latest models optimized for long-context scenarios with RULER can give us more insights. As such, we provide additional evaluations of Gemini-1.5-pro, Command-R-plus, Mixtral-8x22B, Phi3, DBRX, Qwen along with results from our submitted version in the table below. We plan to evaluate Claude 3 and Moonshot after finalizing the evaluation pipeline of these two models.
>
> |Models|Claimed Length|Effective Length|4K|8K|16K|32K|64K|128K|Avg.|wAvg. (inc)|wAvg. (dec)|
> |:---:|:---:|:---:|:---:|:---:|:---:|:---:|:---:|:---:|:---:|:---:|:---:|
> |Llama2 (7B)|4K||85.6|
> Gemini-1.5-pro1M | 128K| >128K|96.7|95.8| 96.0| 95.9|95.9|94.4|95.8|95.5 (1st)|96.1 (1st)|
> GPT-4-1106-preview|128K|64K|96.6|96.3| 95.2| 93.2|87.0|81.2|91.6|89.0 (2nd)|94.1 (2nd)|
> Command-R-plus(104B)|128K|32K| 95.6| 95.2| 94.2|92.0|84.3|63.1|87.4|82.7 **(5th)**|92.1 **(3rd)**|
> Command-R (35B)|128K|32K| 93.8| 93.3| 92.4|89.5|84.9|76.0|88.3|85.5 **(3rd)**|91.1 **(4th)**|
> Mixtral-8x22B (39B/141B)|64K|32K| 95.6| 94.9| 93.4|90.9|84.7|31.7|81.9|73.5 **(6th)**|90.3 **(5th)**|
> Yi (34B)|200K|32K| 93.3| 92.2| 91.3|87.5|83.2|77.3|87.5|84.8 **(4th)**|90.1 **(6th)**|
> Mixtral-8x7B (12.9B/46.7B)|32K|32K| 94.9| 92.1| 92.5|85.9|72.4|44.5|80.4|72.8 (7th)|87.9 (7th)|
> Mistral (7B)|32K|16K| 93.6| 91.2|87.2|75.4|49.0|13.8|68.4|55.6 **(11th)**|81.2 **(8th)**|
> ChatGLM (6B)|128K|4K|87.8|83.4|78.6|69.9|56.0|42.0|69.6|62.0 (10th)|77.2 (9th)|
> LWM (7B)|1M|<4K|82.3|78.4|73.7|69.1|68.1|65.0|72.8|69.9 **(8th)**|75.7 **(10th)**|
> Phi3 (3.8B)|128K|4K|86.7|78.1|75.6|70.3|58.9|43.3|68.8|62.2 **(9th)**|75.5 **(11th)**|
> DBRX (36B/132B)|32K|8K|95.1|93.8|83.6|63.1|2.4|0.0|56.3|38.0 (12th)|74.7 (12th)|
> Qwen (72B)|32K|8K|94.9|93.8|78.0|67.8|0.0|0.0|55.7|37.5 (13th)|74.0 (13th)|
> Together (7B)|32K|4K|88.2|81.1|69.4|63.0|0.0|0.0|50.3|33.8 (14th)|66.7 (14th)|
> LongChat (7B)|32K|<4K|84.7|79.9|70.8|59.3|0.0|0.0|49.1|33.1 (15th)|65.2 (15th)|
> LongAlpaca (13B)| 32K|<4K|60.6|57.0|56.6|43.6|0.0|0.0|36.3|24.7 (16th)|47.9 (16th)|

---

### Decision · Program_Chairs · 2024-07-10

**Decision:**

Accept

**Comment:**

The authors propose a novel synthetic benchmark called RULER for evaluating the long-context processing abilities of LLMs. Unlike the "needle in a haystack" benchmark, which is too easy, RULER consists of more comprehensive and challenging tasks, including multi-hop tracing and aggregation. The findings are straightforward: most LLMs are not fully functional at their nominal context size, but a few perform well at a context size of 32k. Even though some models support larger context sizes, they struggle with handling those large inputs.

The reviewers found this work:

- Provides extensive and thorough evaluations of LLMs on diverse tasks.
- Offers detailed analysis.
- Is well written.
- Proposes a benchmark and findings that will be helpful for LLM researchers and practitioners.

Overall, we recommend acceptance.